# Open-ended VQA benchmarking of Vision-Language models by exploiting Classification datasets and their semantic hierarchy

**Simon Ging**\*  **María A. Bravo**\*  **Thomas Brox**
{gings,bravoma,brox}@cs.uni-freiburg.de
University of Freiburg, Germany
https://github.com/lmb-freiburg/ovqa

## Abstract

The evaluation of text-generative vision-language models is a challenging yet crucial endeavor. By addressing the limitations of existing Visual Question Answering (VQA) benchmarks and proposing innovative evaluation methodologies, our research seeks to advance our understanding of these models' capabilities. We propose a novel VQA benchmark based on well-known visual classification datasets which allows a granular evaluation of text-generative vision-language models and their comparison with discriminative vision-language models. To improve the assessment of coarse answers on fine-grained classification tasks, we suggest using the semantic hierarchy of the label space to ask automatically generated follow-up questions about the ground-truth category. Finally, we compare traditional NLP and LLM-based metrics for the problem of evaluating model predictions given ground-truth answers. We perform a human evaluation study upon which we base our decision on the final metric. We apply our benchmark to a suite of vision-language models and show a detailed comparison of their abilities on object, action, and attribute classification. Our contributions aim to lay the foundation for more precise and meaningful assessments, facilitating targeted progress in the exciting field of vision-language modeling.

## 1 Introduction

Image-text generative models have advanced artificial intelligence by connecting visual and language domains. Recently, such models have been scaled up successfully in terms of model size and training data, showing impressive capabilities across various applications. One such capability is open-ended Visual Question Answering (oVQA), which tests vision-language models (VLMs) on their visual understanding by asking questions via natural language. Unlike multiple-choice VQA, where answers can be chosen from a predefined set of options, oVQA requires the model to generate the answer rather than simply choosing the option with the highest score. We refer to such text-generative VLMs as Text-VLMs.

The unconstrained nature of oVQA presents a significant challenge during evaluation. Firstly, natural language has a wide variety of possibilities to express the same content. There exists an inherent difficulty in comparing the semantic similarity of two language expressions, rendering it hard to say whether a model's answer is right or wrong. Current automatic metrics do not capture the richness of model responses, especially when multiple valid answers, synonyms, paraphrasing, or nuanced explanations are possible (Chen et al., 2020; Khashabi et al., 2021). They especially penalize longer responses, leading to a bias towards methods that produce shorter answers (Risch et al., 2021).

Secondly, there is typically ambiguity in a question, which depends on the context provided and the expected scope of the response, resulting in multiple valid answers that may not match the annotated ground truth. For example consider, when asking about the type of animal in a picture, a model may answer *dog*, but the ground truth expects the more detailed *newfoundland dog* as the answer. A natural way of solving this ambiguity is by following up with the model's first response

---

\*These authors contributed equally to this work.

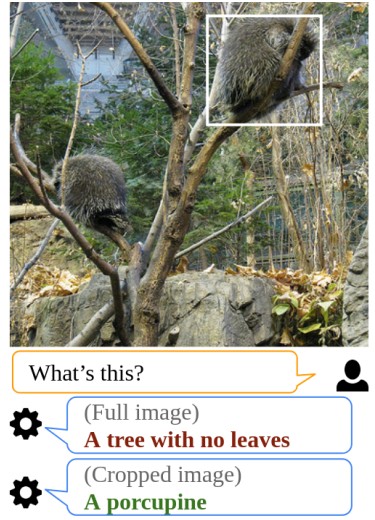

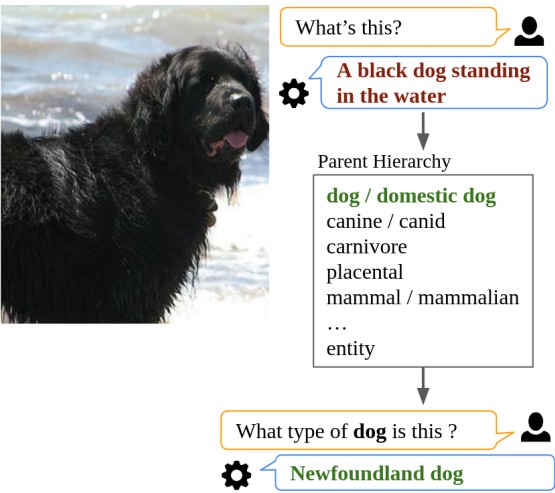

(a) Cropping the biggest object guides the model to the location of the question.

(b) Asking a follow-up question using the concept hierarchy helps to resolve ambiguities.

Figure 1: Our proposed framework for building an open-ended VQA benchmark using visual guidance for question context and probing for details with follow-up questions.

and asking what type of dog it is. This follow-up mechanism is closely related to Image-text dialog in which systems engage in conversations with users about images (Das et al., 2017). It allows interleaving text and image inputs to create visual conversations and enables in-context few-shot learning (Alayrac et al., 2022). However, these systems currently do not seek clarification about missing information in the conversation, as a human would do when a question is unclear.

Finally, current VQA benchmarks only provide a single aggregated performance metric as output and a coarse division into question and answer types based on heuristics. While this format is convenient for comparing different methods, aggregated metrics conceal information about the true strengths and limitations of models. This problem is amplified by biases in the distribution of question-answer pairs that lead methods to use simple statistical shortcuts, e.g. image-blind algorithms often perform well in VQA (Kafle et al., 2019). As a result, classical VQA benchmarks have reached a certain saturation where models achieve high overall performances, with minor differences between them. Nonetheless, we find that models still have clear deficits, as seen in a qualitative inspection of the results, especially in out-of-distribution scenarios. These deficits are hard to explain with a single performance metric, and qualitative comparison on thousands of images is impractical.

In this work, we introduce ways to approach these challenges. To provide more details about the performance of a model on different aspects of VQA, we introduce dedicated subbenchmarks for questions about objects, actions, and attributes, corresponding to nouns, verbs, and adjectives. To this end, we leverage existing classification benchmarks, from which we automatically generate questions and ground truth answers. Additionally, this allows us to compare Text-VLMs with discriminative VLMs that directly classify the image without generating text.

We propose a follow-up procedure to reduce question ambiguity and guide models toward the level of detail expected by the ground truth. When adapting classification benchmarks for oVQA, we address this ambiguity by initially cropping the image to provide the model with the appropriate visual context for the target. Subsequently, after receiving an initial answer from the model, we generate a follow-up question using concept hierarchies to enable the model to generate an answer with the desired granularity.

To select an appropriate metric for evaluation, we compare various metrics from the VQA and NLP communities that consider paraphrasing and synonyms as correct answers. We validate the selected metrics by a human-judgment study of the answers as a gold standard metric. Finally, we evaluate several state-of-the-art VLMs on our extended VQA benchmark and provide an in-depth analysis of the strengths and weaknesses identified.

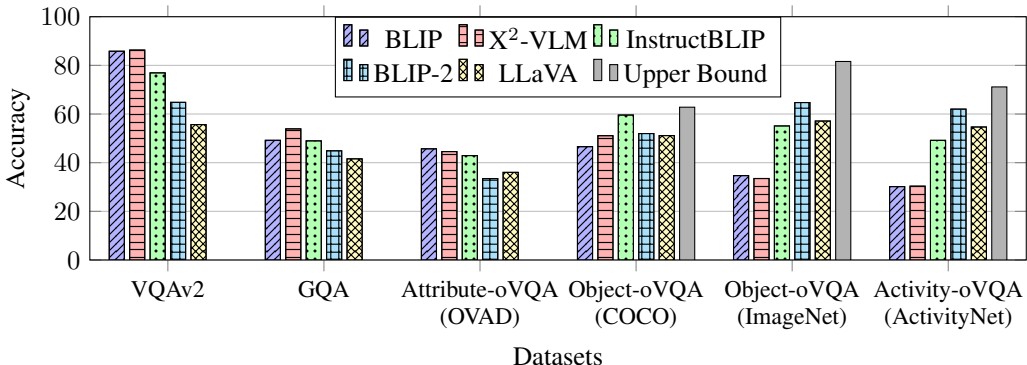

Figure 2: Comparison of Models on open-ended VQA datasets, testing traditional VQA, Object-oVQA, Activity-oVQA, and Attribute-oVQA. Generic pretrained models like BLIP-2 are stronger on predicting objects and activities with high semantic granularity while exhibiting lower performances for Attribute-oVQA and classical VQA tasks. Contrary, VQA pretrained models are stronger on predicting attribute concepts and answering generic VQA questions but perform poorly for fine-granularity of nouns and activities.

## 2 RELATED WORK

Large VLMs have substantially advanced in various multimodal tasks, including visual question answering and multimodal conversation. Recent studies have introduced more efficient vision-text interactions, enhanced training techniques, and incorporated instruction tuning into VLMs, such as Llama-adapter v2 (Gao et al., 2023) and LLaVA (Liu et al., 2023). Other notable VLMs include Flamingo (Alayrac et al., 2022), which integrates visual features into Large Language Models (LLMs) through cross-attention layers, and BLIP-2 (Li et al., 2023b), which pre-trains a module for zero-shot image-to-text generation, achieving improved performance in image captioning and VQA.

Evaluating the performance of such VLMs poses a significant challenge, with no uniform standard for testing. For instance, Liu et al. (2023) assess VLMs by generating predictions for a few manually curated image-question pairs and comparing them to a reference prediction generated by GPT-4. Bai et al. (2023) focus on evaluating conversational and visual storytelling abilities, employing a manually created visual dialogue dataset. Their evaluation methodology compares model outputs to detailed image annotations, revealing insights into model performance relative to human judgments. However, these evaluation criteria are not standardized. A quantitative comparison between a pretrained Text-VLM like BLIP-2, instruction-tuned models like LLaVA, and retrieval models like CLIP is not feasible since benchmarks are only applicable to the respective tasks designed for such types of models. Improvements in this area would ease model selection for applications and enable more targeted research to overcome models' previous weaknesses.

A great interest has been shown in the VQA task. Antol et al. (2015) create the first free-form and open-ended Visual Question Answering dataset by asking human workers to create questions that a smart robot probably can not answer and then collect ten human answers per question. The follow-up work VQAv2 (Goyal et al., 2017) balances the previous dataset to reduce statistical biases in the answer distribution. Opposed to leveraging human workers, the GQA dataset (Hudson & Manning, 2019) uses scene graph structures from Visual Genome (Krishna et al., 2017) to generate question-answer pairs. These graph structures enable the authors to balance the answer distributions for each question, reducing the efficacy of depending on answer statistics. Both datasets remain widely used today (Dai et al., 2023; Zeng et al., 2022; Alayrac et al., 2022).

In contrast to Text-VLMs, assessing LLMs has been a focal point of research within the Natural Language Processing (NLP) community. Diverse methods, including LLaMA (Touvron et al., 2023a), GPT-3 (Brown et al., 2020), Chinchilla (Hoffmann et al., 2022), and PaLM (Chowdhery et al., 2022), are trained on large amounts of text and evaluated in zero-shot or few-shot scenarios. These evaluations traverse commonsense reasoning, question answering, reading comprehension, mathematical reasoning, and code generation. Comparing candidate and reference text has been

extensively studied in the context of machine translation (Mathur et al., 2020; Kocmi et al., 2021) and image captioning (Cui et al., 2018). Traditional automated metrics like BLEU (Papineni et al., 2002), ROUGE (Lin, 2004), METEOR (Banerjee & Lavie, 2005), CIDEr (Vedantam et al., 2015) and SPICE (Anderson et al., 2016) aim to approximate human judgments, using heuristics in the text space. Some follow-up works like BLEURT (Sellam et al., 2020), LERC (Chen et al., 2020), and BEM (Bulian et al., 2022) propose learning metrics in a supervised manner on human ratings, while others like BertScore (Zhang et al., 2020) use feature spaces from transformers to compare text. However, these evaluation efforts mostly apply to comparing longer sentences. As we show in this work, they do not improve over simple text comparison when presented with very short ground truth answers and mixed length model predictions.

A recent focus has been assessing LLMs' capabilities in human-like dialogues like chatbots. Chiang et al. (2023) evaluate Vicuna, their chatbot, by pinning it against another assistant and having GPT-4 compare and rank both model outputs. Zheng et al. (2023) show the agreement between GPT-4 and humans to be comparable to the inter-human agreement on their MT-bench data, which consists of 80 multi-turn questions. Despite these extensive evaluation efforts, challenges such as overfitting to benchmarks and training distribution biases, capturing correctly a semantic similarity between synonyms and paraphrases remain partially addressed.

## 3 OPEN-ENDED VQA BENCHMARK

### 3.1 TRANSFORMING CLASSIFICATION DATASETS TO VQA

The main goal of this work is to assess Text-VLMs' performance in an open-ended VQA (oVQA) setup. To achieve this, we transform traditional image classification datasets into a VQA format to create new, fine-grained VQA subbenchmarks. We consider two types of image classification datasets, multi-class classification and binary classification, to build our oVQA benchmark. Supp. Fig. 9 shows an example for the 4 proposed oVQA datasets and 2 classical VQA datasets.

**Multi-class classification.** In multi-class classification, the goal is to predict the correct class index $t \in \{1, \ldots, C\}$ from a set of $C$ classes, given an input image $I$. Similar to other works that test VLMs for classification, such as Radford et al. (2021), we use the class names $c$ to enable zero-shot classification. For every image-class pair $(I, c)$, we generate a question such that Text-VLM can produce the corresponding class name.

**Binary classification.** In binary classification, the task is to predict whether specific categories $c$ apply to each image or not. To address this task in oVQA, we consider only the positive categories for each image $I$ to generate relevant questions for the Text-VLM.

**Visual guidance for question context.** Consider the example depicted in Fig. 1a from the ImageNet dataset. Imagine you want a Text-VLM to identify the ImageNet category by asking a very general question, such as "What's this?" Unfortunately, the task becomes ambiguous if given such a broad question and the entire image as context. A Text-VLM might provide a very general response, such as "a tree with no leaves." Although it is not a wrong prediction, it is incorrect given the "porcupine" label. To resolve this ambiguity, we crop the object of interest given the ground-truth box.

**Metrics.** To evaluate the correctness of an answer, it is required to compare a model's output with a class name. We begin with the simplest evaluation metric, called *ExactMatch* (*EM*). It involves preprocessing the model's output $p$, a free-format text that resembles natural language with human-like fluency, and the corresponding class name $c$ by removing non-alphanumeric characters, applying lowercase conversion, and normalizing spaces. For *EM* a prediction is considered correct only if the model's output exactly matches the correct class name. A less restrictive option is to consider a response correct if the prediction contains the true class name after preprocessing (Xu et al., 2023), named as *Contains* (*Cont*). We use both metrics to evaluate Text-VLMs and extend both metrics in the presence of category synonyms, considering the prediction as correct if any one of the synonyms is correct under each of the metrics. We denote these extensions as *EM Syn* and *Cont Syn*.

For multi-class classification, an image belongs to only one of the categories $C$. Therefore, we adopt the evaluation introduced by Conti et al. (2023). We report matching the prediction and label using cosine similarity in a vector embedding space and name this metric as *ClipMatch* (*ClipM*). For this, we embed the model prediction and each class name using EVA-Clip (Fang et al., 2023).

Subsequently, we consider the prediction as correct if the vector corresponding to the correct class exhibits the highest similarity (*ClipM@1*) or is within the top-5 similarities (*ClipM@5*).

**Probing for details with a follow-up question.** The expected specificity of the task is typically unclear when asking in an open-ended way. Take, for example, Fig. 1b. When asking "What's this?", the model might respond with "A black dog standing in the water," which is an accurate answer, but not the specific category in the ground truth. Therefore, we propose a follow-up procedure to enhance the precision of model responses in oVQA, especially for fine-grained questions. We assume a hierarchy of parents for each class as shown in Fig. 1b. First, we consider all data points where the model response is judged as incorrect by the *ClipM@1* metric. Then, we collect all parent nodes in the hierarchy of the ground truth label and calculate their similarity to the model prediction using EVA-Clip. We define the parent with the highest similarity as the candidate for the follow-up. In the example, we compare the prediction and the parent class of "Newfoundland dog". We find the prediction to be sufficiently close to the parent "dog" and ask the follow-up question "What type of dog is this?"

Our goal is to only ask follow-up questions about concepts that the model already described. Therefore, we only ask a follow-up question about the parent if the similarity with the prediction is at least $\delta$. When the similarity is lower than $\delta$, we instead define the parent as a generic term like "object" or "activity", depending on the task, in order to avoid disclosing part of the ground truth information. This systematic procedure, rooted in lexical and semantic similarity assessment, refines the model's responses in oVQA, enhancing precision and contextuality.

**Upper bound.** In order to compare Text-VLMs with discriminative VLMs and infer how far away they are from each other, we evaluate discriminative VLMs in the same multi-class classification VQA datasets using zero-shot retrieval. It is important to note that generating the correct class name given the visual input in the language space is considerably harder than generating a good visual embedding vector representing the visual concepts. Similar to Conti et al. (2023), we consider the retrieval performance as an upper bound for our evaluation.

## 3.2 Datasets and experimental setup

**Object-oVQA.** For the task of object classification oVQA, we consider two well-known datasets: MS COCO for the common categories with coarse granularity and ImageNet for the fine granularity.

The COCO (Common Objects in Context) (Lin et al., 2014) dataset is a widely used dataset designed for object recognition. It contains 80 object categories easily recognizable by a 4-year-old, such as "dog", "person", "boat", etc. We crop each of the 36,781 objects from the validation 2017 set and consider them individually. Since the object categories already have a coarse granularity, we do not ask a follow-up question for the COCO-oVQA task.

ImageNet (Deng et al., 2009) is a standard benchmark for image classification (Russakovsky et al., 2015). Its validation set contains 50000 images from 1000 classes, including simple labels such as "broom" and fine-grained labels like "Lakeland terrier" and 22 other types of "terrier". For each image, we crop the biggest bounding box. We define follow-up questions using the WordNet (Miller, 1994) hierarchy and ask "What type of *object* is this?" by replacing "object" with the closest parent category.

**Activity-oVQA.** We use the ActivityNet (Caba Heilbron et al., 2015) dataset and the activity category hierarchy provided by the original paper. It consists of 4,926 untrimmed validation videos belonging to 200 activity classes. To build the ActivityNet-oVQA task, we use the annotated video segments (in which the activity is happening) from the validation set and extract the middle frame. This way, we obtain 7,654 frames within 200 activity categories in a multi-class classification setup. This dataset is considered as having fine-grained categories since the activity categories are very specific. Therefore, we follow a similar procedure as in ImageNet-oVQA: Initially, we ask the model a general question and create follow-up questions when the first answer is not hitting the exact class label. We select the parent node most similar to the model's prediction to form the follow-up question. Specifically, we ask, "What type of *activity* is this?" by replacing the word "*activity*" with the closest parent category.

**Attribute-oVQA.** For the attribute classification, we use the OVAD dataset, introduced by Bravo et al. (2023), which focuses on common adjective concepts like color, material, length, and others.

Table 1: Zero-shot retrieval results for the evaluated multi-class classification datasets using discriminative VLMs on ImageNet (cropped), COCO objects and ActivityNet.

| Model | INet-Crop | | COCO obj | | ActivityNet | |
|---|---|---|---|---|---|---|
| | R@1 | R@5 | R@1 | R@5 | R@1 | R@5 |
| CLIP ViT-L-14 | 77.90 | 95.30 | 55.20 | 85.35 | 68.76 | 89.27 |
| CLIP EVA 01-g-14 | **81.60** | **96.62** | 62.81 | **89.67** | **71.14** | **89.98** |
| BLIP-2 | 69.20 | 89.70 | 57.16 | 89.04 | 61.03 | 82.27 |
| BLIP-2 (COCO ft) | 65.21 | 87.05 | **62.98** | 89.04 | 59.59 | 81.36 |

Table 2: OVAD-oVQA (cropped object bounding boxes) accuracy. Results over three different questions depending on the attribute type. See Supp. Fig. 3 for the specific questions.

| Method | Cont Syn $\pm$ | | EM Syn $\pm$ | |
|---|---|---|---|---|
| BLIP$_{vqa}$ | **45.70** | 5.46 | 36.99 | 5.00 |
| X$^2$-VLM$_{vqa}$ B | 43.01 | 7.24 | 37.26 | 6.68 |
| X$^2$-VLM$_{vqa}$ L | 44.56 | 6.33 | 39.13 | 5.62 |
| BLIP-2 OPT | 21.89 | 1.41 | 5.23 | 0.52 |
| BLIP-2 T5 | 33.45 | 0.94 | 15.73 | 3.69 |
| InstructBLIP T5 | 43.29 | 0.53 | 38.88 | 2.28 |
| InstructBLIP V | 42.92 | 6.33 | **40.42** | 5.81 |
| LLaVA | 36.02 | 1.29 | 0.00 | 0.00 |

The OVAD dataset contains object-level attribute annotations for 117 attribute categories and 80 object classes over 2000 images. Attributes are organized in a two-level hierarchy such that every attribute is under a parent attribute type. We crop every object using the ground-truth bounding box as in COCO-oVQA and consider it an independent visual input. To formulate questions for the VQA task, we employ the parent attribute types within the hierarchy to inquire about specific attributes effectively. For instance, if the ground-truth annotation of a person specifies "green" under the "clothes color" attribute type, we construct the question, "What colors are the clothes the person is wearing?". A complete list of questions is included in the supplementary. We employ this approach to generate questions for each positive attribute annotated in the OVAD dataset. The final OVAD-oVQA dataset consists of 122,997 object-attribute-question tuples. During evaluation, we consider the synonyms for every positive attribute category as correct answers. Since the OVAD-oVQA consists of a binary classification problem, we employ the metrics *EM Syn* and *Cont Syn*.

**Classical VQA.** To link traditional VQA evaluation with our proposed benchmark, we evaluate on two existing VQA datasets. The VQAv2 dataset (Goyal et al., 2017) aims to test systems for detailed image understanding and complex reasoning. We evaluate the validation split with 214K questions and the test-dev split with 104K questions. For test-dev, the ground truth answers are reserved, and evaluation is done by submitting model predictions online. For each question, the dataset contains 10 human-annotated answers. We follow the authors and calculate our metrics as follows. Given model prediction $P$ and human answers $h_i, i \in 1, \ldots, 10$, the final score for the prediction is $\min(1, 0.3 \sum_i \text{metric}(P, h_i))$ e.g. for *ExactMatch* (*EM*), if 3 humans agree with the model, the score is 90%. We utilize the same text preprocessing as the VQAv2 dataset work. The GQA dataset (Hudson & Manning, 2019) uses scene graph structures from Visual Genome (Krishna et al., 2017) to generate question-answer pairs. Evaluating all models on the "balanced-testdev" split, our analysis encompasses 13k questions across 400 images. The preprocessing mirrors that of the VQAv2 dataset. Opposed to VQAv2, only one ground truth answer exists per question, so we report *EM* and *Cont* metrics between this answer and the model prediction.

## 3.3 TESTED VLMs

We compare state-of-the-art models in classical VQA and recent conversational models. We selected two models fine-tuned on VQAv2, BLIP (Li et al., 2022) and X$^2$-VLM (Zeng et al., 2022); one multi-purpose model evaluated in a zero-shot manner BLIP-2 (Li et al., 2023b); and two conversational models InstructBLIP (Dai et al., 2023) and LLaVA (Liu et al., 2023). All models consist of an image encoder and text encoder based on the transformer architecture as well as a fusion model. They use pretrained models for the unimodal encoders and train the fusion model using image-text data. BLIP and X$^2$-VLM fine-tune the encoders together with the fusion module using common losses such as Image-Text Contrastive Loss (ITC), Image-Text Matching Loss (ITM), and image-conditional Language Modeling Loss (LM). BLIP-2, InstructBLIP, and LLaVA keep the unimodal encoders frozen and only train the fusion module. Part of the training data of InstructBLIP includes the VQAv2 dataset. We report the models' details in the Supp. Appendix A.1.

Table 3: ImageNet-oVQA (cropped bounding box) object classification accuracy. **Top:** Results for asking the questions "What can be seen in the image?", "What is in the image?" and "What's this?" **Bottom:** Results for asking follow-up question - "What type of *object* is this?" as a response to the output of the first question. We highlight the **first**, second and *third* best method.

| | Method | ClipM@1 $\pm$ | | ClipM@5 $\pm$ | | Cont $\pm$ | | EM $\pm$ | |
|---|---|---|---|---|---|---|---|---|---|
| | BLIP (vqa finetuned) | 24.51 | 0.95 | 43.83 | 1.70 | 10.22 | 0.85 | *9.24* | 1.29 |
| | $X^2$-VLM$_{vqa}$ B | 21.17 | 1.83 | 40.55 | 2.89 | 8.24 | 1.42 | 7.80 | 1.57 |
| | $X^2$-VLM$_{vqa}$ L | 23.67 | 2.29 | 43.64 | 3.40 | 9.42 | 1.59 | 8.84 | 1.80 |
| 1$^{st}$ Question | BLIP-2 OPT | **57.10** | 2.08 | **77.24** | 2.03 | **35.49** | 2.04 | 0.87 | 0.37 |
| | BLIP-2 T5 | 54.50 | 2.35 | 75.71 | 2.26 | 30.05 | 2.89 | 1.77 | 1.59 |
| | InstructBLIP T5 | 41.12 | 3.76 | 60.90 | 3.86 | 20.87 | 2.81 | **19.59** | 3.02 |
| | InstructBLIP V | 39.69 | 3.34 | 59.59 | 3.59 | 20.03 | 2.90 | 19.30 | 2.82 |
| | LLaVA | *45.07* | 0.76 | *72.11* | 0.87 | *23.18* | 0.63 | 0.00 | 0.00 |
| | BLIP (vqa finetuned) | 34.68 | 0.20 | 53.06 | 0.17 | 15.00 | 0.47 | *13.85* | 0.92 |
| | $X^2$-VLM$_{vqa}$ B | 29.24 | 0.13 | 47.43 | 0.01 | 11.64 | 0.52 | 11.07 | 0.69 |
| | $X^2$-VLM$_{vqa}$ L | 33.50 | 0.37 | 52.54 | 0.16 | 13.69 | 0.65 | 12.92 | 0.87 |
| 1$^{st}$ Question | BLIP-2 OPT | **67.22** | 0.33 | **83.54** | 0.14 | **40.31** | 0.90 | 2.54 | 0.27 |
| | BLIP-2 T5 | 64.71 | 0.43 | 80.83 | 0.20 | 34.43 | 1.49 | 4.81 | 0.62 |
| | InstructBLIP T5 | 55.41 | 0.72 | 73.34 | 0.34 | *27.92* | 1.13 | **26.48** | 1.42 |
| | InstructBLIP V | 55.10 | 1.27 | 73.11 | 0.68 | 27.38 | 1.78 | 26.40 | 1.73 |
| | LLaVA | *57.12* | 0.29 | *79.13* | 0.03 | 27.59 | 0.31 | 0.00 | 0.00 |

# 4 RESULTS

We evaluate the models from Sec. 3.3 on our benchmark, reporting the performance on objects, activities, attributes, and classical VQA. Fig. 2 summarizes the results. To reduce the dependency on a single prompt, we ask three questions for each visual input and report the mean and standard deviation.

**Retrieval Upper Bound.** Tab. 1 shows the upper bound performance, i.e., when all allowed classes are tested explicitly by a prompt and the maximum score yields the decision (multiple-choice setting). Recent works like EVA improve over the original CLIP on object and action classification. Despite impressive results on caption retrieval (Li et al., 2023b), BLIP-2 performs worse than CLIP on fine-grained (ImageNet) object retrieval but comparable on coarse classes (COCO obj).

**ImageNet-oVQA.** We report the performance for every model in Tab. 3 by first asking a general question (upper half) and then a more specific follow-up question (lower half). We observe that BLIP-2 outperforms both instruction-tuned models like InstructBLIP and LLaVA, as well as VQA-finetuned models. These results highlight that models pretrained on large, generic data collections perform well in a zero-shot task with fine-grained classes. Additionally, we illustrate the effectiveness of our experimental framework in the Supplementary (see Supp. Fig. 4). In terms of metrics, we find that *ClipM* is able to rank the correct label as most similar among possible labels, even if the prediction does not contain the exact label text. For example, using *ClipM*, the model answer "a mountain lion" is marked as correct; however, it does not match the label "cougar" under the metrics *EM* and *Cont* (see Supp. Fig. 4a). We observe that follow-up questions allow for a fair comparison between models and improve the performance by 32% on average across tested models. See Supp. Fig. 4b for qualitative examples.

**COCO-oVQA.** For this task, we show the results in Tab. 4. InstructBlip models show a higher performance for the *ClipM* and *EM* metrics with a high sensitivity *w.r.t.* the question being asked (high standard deviation). The method produces very short answers in most of the cases. Other methods like LLaVA and BLIP-2 show a competitive performance when measuring *ClipM* metric but are highly penalized by *EM* mainly due to producing long answers. As compared to any of the other datasets, the gap to the retrieval upper bound in Tab. 1 is much closer, showing that for a coarse granularity of categories Text-VLMs are reasonably good classifiers. See Supp. Fig. 6 for some qualitative examples.

**ActivityNet-oVQA.** Similar to ImageNet-oVQA, we evaluate models for activity classification (Tab. 5) in two setups: First asking a general question (upper half) and then posing a more specific

Table 4: COCO-oVQA (cropped bounding box) object classification accuracy. Results for asking the questions "What can be seen in the image?", "What is in the image?" and "What's this?"

| Method | ClipM@1 | $\pm$ | ClipM@5 | $\pm$ | EM | $\pm$ | Cont | $\pm$ |
|---|---|---|---|---|---|---|---|---|
| BLIP$_{vqa}$ | 46.58 | 3.35 | 65.54 | 4.05 | 21.74 | 3.87 | 26.04 | 1.11 |
| X$^2$-VLM$_{vqa}$ B | 48.22 | 7.46 | 63.72 | 5.51 | 23.53 | 1.23 | 25.76 | 1.34 |
| X$^2$-VLM$_{vqa}$ L | 51.09 | 4.79 | 67.82 | 2.50 | 22.50 | 2.41 | 25.00 | 1.30 |
| BLIP-2 OPT | 47.62 | 0.90 | 70.93 | 4.76 | 0.46 | 0.19 | 33.59 | 3.20 |
| BLIP-2 T5 | 51.97 | 2.25 | 76.21 | 3.10 | 0.10 | 0.14 | 32.50 | 0.86 |
| InstructBLIP T5 | 59.38 | 5.88 | 72.80 | 3.61 | 25.07 | 1.55 | 26.32 | 1.77 |
| InstructBLIP V | **59.58** | 8.22 | 73.32 | 4.92 | **26.50** | 3.56 | 27.52 | 3.96 |
| LLaVA | 51.13 | 1.17 | **81.04** | 0.32 | 0.00 | 0.00 | **45.08** | 2.33 |

Table 5: ActivityNet-oVQA (center frame) activity classification accuracy. **Top:** Results for asking the questions "What activity is this?", "What is happening in the image?" and "What is this?" **Bottom:** Results for asking follow-up question: "What type of *activity* is this?"

| | Method | ClipM@1 | $\pm$ | ClipM@5 | $\pm$ | EM | $\pm$ | Cont | $\pm$ |
|---|---|---|---|---|---|---|---|---|---|
| 1$^{st}$ Question | BLIP$_{vqa}$ | 23.73 | 1.28 | 43.04 | 1.98 | 6.70 | 4.92 | 7.30 | 4.70 |
| | X$^2$-VLM$_{vqa}$ B | 21.02 | 1.59 | 42.27 | 3.56 | 5.41 | 4.18 | 5.86 | 4.14 |
| | X$^2$-VLM$_{vqa}$ L | 24.21 | 2.08 | 46.27 | 3.73 | 6.34 | 4.17 | 6.92 | 4.00 |
| | BLIP-2 OPT | **55.08** | 0.35 | 74.32 | 0.70 | 7.10 | 7.28 | 14.66 | 5.03 |
| | BLIP-2 T5 | 53.39 | 3.02 | **74.71** | 2.59 | 7.07 | 9.59 | **15.70** | 3.78 |
| | InstructBLIP T5 | 40.18 | 0.71 | 61.82 | 0.92 | 10.87 | 3.83 | 11.64 | 4.10 |
| | InstructBLIP V | 40.64 | 2.67 | 62.35 | 3.41 | **12.51** | 1.87 | 13.61 | 2.95 |
| | LLaVA | 46.75 | 0.47 | 72.26 | 0.61 | 0.00 | 0.00 | 8.76 | 1.08 |
| Follow-up Question | BLIP$_{vqa}$ | 30.16 | 2.28 | 44.14 | 2.48 | 8.05 | 5.04 | 8.65 | 4.86 |
| | X$^2$-VLM$_{vqa}$ B | 25.74 | 2.91 | 42.17 | 4.42 | 6.52 | 4.01 | 6.92 | 4.00 |
| | X$^2$-VLM$_{vqa}$ L | 30.35 | 2.94 | 45.64 | 4.48 | 7.65 | 3.89 | 8.24 | 3.76 |
| | BLIP-2 OPT | 60.96 | 1.28 | 73.90 | 0.96 | 7.68 | 6.93 | 16.14 | 4.54 |
| | BLIP-2 T5 | **62.02** | 2.61 | **75.13** | 0.95 | 8.84 | 8.99 | **18.09** | 3.36 |
| | InstructBLIP T5 | 50.23 | 2.84 | 66.11 | 2.95 | 13.31 | 4.56 | 14.08 | 4.89 |
| | InstructBLIP V | 49.18 | 3.49 | 66.34 | 3.26 | **13.99** | 1.99 | 15.09 | 3.14 |
| | LLaVA | 54.67 | 1.68 | 73.94 | 0.63 | 0.00 | 0.00 | 10.68 | 1.08 |

follow-up question (lower half). Activity questions may be the most out-of-distribution questions with respect to the classical VQA datasets, which might be the reason for the very poor performance of finetuned models on VQA (BLIP, X$^2$-VLM). On this task, BLIP-2 models outperform all other models for both *ClipM* and *Cont* metrics. Follow-up questions consistently improve the performance of all models but remain far from the upper bound (Tab. 1). See Supp. Fig. 5 for some qualitative examples.

**OVAD-oVQA.** For the attribute classification task (see Tab. 2), we observe a similar behavior as in the classical VQA case (Tab. 6): Models finetuned or instruction-tuned on VQA (BLIP, X$^2$-VLM, InstructBLIP) perform well, while BLIP-2 and LLaVA show a very poor performance. This is likely due to the imbalanced pretraining dataset where attributes often lie in the tail of the distribution, skewing such models more towards nouns and verbs. From qualitative analysis, we observe that the BLIP-2 OPT model tends to answer with objects regardless of the attribute-related question, and LLaVA hallucinates long answers, ignoring the image when asked some contextual questions like the number of objects in the image. These could potentially explain their lower performance on the OVAD-oVQA task. See Supp. Fig. 6 for some qualitative examples.

**Classical VQA.** Tab. 6 shows the results on classical VQA datasets. Methods finetuned on VQAv2 outperform zero-shot models on both VQA datasets; however, the zero-shot methods are reasonably competitive. We find that *EM* is unsuitable for evaluating instruction-tuned models like LLaVA, reporting 0% accuracy in Tab. 6 even for reasonable answers (illustrated in Supp. Fig. 8a). *Cont* is a better choice to evaluate such models. This finding aligns with the results from the user study described in the following paragraph. However, in some cases *Cont* may overestimate model per-

Table 6: Classical VQA. Zero-shot methods (top) and models finetuned on VQA (bottom). $^g$: Authors use rank-based evaluation with a fixed vocabulary, we use generative evaluation. *Italic*: Evaluated data is part of training.

| Model | VQAv2 | | | GQA testdev | |
|---|---|---|---|---|---|
| | test-dev | val | | balanced | |
| | EM | EM | Cont | EM | Cont |
| BLIP-2 OPT | 51.32 | 51.51 | 52.66 | 32.21 | 32.93 |
| BLIP-2 T5 | **62.80** | **63.13** | **64.80** | **44.03** | **44.88** |
| LLaVA | 0.00 | 0.00 | 55.66 | 0.00 | 41.58 |
| BLIP$_{vqa}$$^g$ | 77.50 | *83.69* | *85.81* | 47.04 | 49.20 |
| X$^2$-VLM$_{vqa}$ B$^g$ | 79.64 | *83.05* | *84.00* | 52.54 | 53.62 |
| X$^2$-VLM$_{vqa}$ L$^g$ | **81.34** | *85.33* | *86.29* | **52.80** | **53.90** |
| InstructBLIP T5 | *74.75* | *73.22* | *74.29* | 47.61 | 48.46 |
| InstructBLIP V | *77.54* | *76.12* | *76.92* | 48.34 | 48.97 |

Table 7: Pearson correlation to human judgement scores on the VQAv2 dataset.

| Metric\Model | Both | BLIP-2 | LLaVA |
|---|---|---|---|
| GPT-4$_{10\text{-}shot}$ | **0.972** | **0.979** | **0.965** |
| GPT-4$_{0\text{-}shot}$ | 0.947 | 0.964 | 0.929 |
| Llama2$_{5\text{-}shot}$ | 0.919 | 0.943 | 0.894 |
| Llama2$_{0\text{-}shot}$ | 0.798 | 0.733 | 0.865 |
| Cont | 0.906 | 0.917 | 0.894 |
| EM | 0.525 | 0.837 | 0.000 |
| BertScore | 0.223 | 0.357 | -0.081 |
| BLEURT-20 | 0.554 | 0.636 | 0.573 |
| BEM | 0.627 | 0.747 | 0.526 |
| LERC | 0.827 | 0.926 | 0.720 |
| BLEU-1 | 0.637 | 0.898 | 0.693 |
| BLEU-4 | 0.023 | 0.027 | 0.014 |
| CIDEr | 0.597 | 0.841 | 0.435 |
| METEOR | 0.689 | 0.895 | 0.830 |
| ROUGE | 0.717 | 0.913 | 0.756 |

formance. Using *Cont*, the model's prediction is deemed as completely correct even if the model predicts several answers, out of which only one is correct (see example in Supp. Fig. 8b).

**User Study: Comparing automatic metrics to human judgment.** In order to evaluate the effectiveness of metrics for comparing a model prediction and ground truth answer, we collect human judgment scores on the VQAv2 dataset. Tab. 7 shows the Pearson correlation between evaluated metrics and our manually curated human gold standard for model predictions from BLIP-2 T5 with short answers and LLaVA with long answers. For more details, we refer the reader to Supp. Appendix C. We find that depending on the budget and size of the evaluation, GPT-4 (OpenAI, 2023a) is a strong choice. LLaMA-2 (Touvron et al., 2023b) is a good option that does not depend on an external service but is still computationally expensive with 70B parameters. For large-scale comparisons, *Cont* is a more reliable choice than *EM* but might pose a risk of degrading in terms of correlation with human judgment when optimizing hyperparameters for it.

## 5 CONCLUSION

In summary, our work has several significant implications: We establish a complete framework for transforming classification benchmarks into VQA benchmarks. By removing ambiguities from the VQA evaluation setup using concept hierarchies to follow up with the model, we enable a comparable and fair evaluation of Text-VLMs on open-ended VQA. Based on this framework, we create a stratified benchmark for diagnosis of Text-VLMs on answering questions about objects, actions, and attributes.

Our user study shows that none of the existing NLP metrics improve VQA evaluation over a simple text comparison metric, with the notable exception of using GPT-4 or LLaMA-2. This finding encourages the use of LLMs as a gold standard metric to reduce the cost of human annotation and motivates research to improve cost-effective Text-QA and VQA metrics. Our selection of metrics enables evaluating Image-dialog models without unfairly penalizing the use of synonyms or longer answers. Together, these contributions enable direct comparison of classical Text-VLMs, Image-dialog models, and discriminative VLMs on the same datasets.

Finally, we show an imbalance in model strengths as highlighted in Fig. 2. Pretrained generic models are stronger in classifying fine-grained objects and activities, while models trained or finetuned for VQA are stronger in classifying coarse concepts and answering common VQA questions, highlighting the possible directions of improvement.

ACKNOWLEDGEMENTS

This work was funded by the German Research Foundation (DFG) - 401269959, 417962828, and 499552394 (SFB 1597 - Small Data). This work was supported by the German Academic Exchange Service (DAAD) - 57440921 Research Grants - Doctoral Programmes in Germany, 2019/20. The authors acknowledge support from the state of Baden-Württemberg through bwHPC. We thank Max Argus and Sudhanshu Mittal for their critical discussions, support, and feedback on the paper. We thank annotators Max Argus, Sudhanshu Mittal, Simon Schrodi, Johannes Dienert, Philipp Schröppel, and Leonhard Sommer for their help, time, and effort.

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

## A  EVALUATION PROTOCOL

### A.1  MODEL DETAILS

In this section and Supp. Tab. 8 we provide additional information about the models we evaluate.

- BLIP (Li et al., 2022) is a vision-language model trained jointly with three objectives: Image-Text Contrastive Loss (ITC) aligning visual and textual feature spaces, Image-Text Matching Loss (ITM), a binary classification loss for positive or negative image-text pairs, and Language Modeling Loss (LM) generating textual descriptions conditioned on images. The pre-trained model is then fine-tuned for various downstream tasks, in particular VQA. Additionally, BLIP generates synthetic captions and filters noisy captions from web-collected image-text pairs to improve data quality and vision-language learning.

- BLIP-2 (Li et al., 2023b) is a VLM that leverages frozen pre-trained image encoders and LLMs by only training an intermediate Q-former, a 12-layer Transformer encoder to join the multimodal data. They used ITC, LM, and ITM losses in the first stage to train the model using only the image encoder and the Q-former. In the second stage, they integrate the pre-trained language model and only use the LM loss. The VQA task is evaluated in a zero-shot manner. When evaluating using the VQA datasets (VQAv2 and GQA) in our work, we use the suggested VQA hyperparameters from the authors: We limit the maximum number of tokens to 10 and set a length penalty of $-1$ during beam search to enforce shorter answers.

- InstructBLIP (Dai et al., 2023) builds an instruction tuning framework based on the pre-trained BLIP-2. It uses 26 publicly available datasets grouped into 11 task categories and introduces instruction-aware visual feature extraction, allowing flexible feature extraction based on given instructions. For the frozen LLM, InstructBLIP uses FlanT5 or Vicuna.

- LLaVA (Liu et al., 2023) is an open-source Large Language-and-Vision Assistant. It connects a pre-trained CLIP ViT-L/14 visual encoder and a large language model LLaMA using a simple projection matrix (linear layer). It considers a two-stage instruction-tuning procedure: In the first stage, the authors pre-train for Feature Alignment and update only the projection matrix based on a subset of the CC3M dataset. In the second stage, the authors fine-tune the projection matrix and LLM end-to-end for multimodal instruction-following data (Visual Chat) and a multimodal reasoning dataset for the science domain (Science QA).

- $X^2$-VLM (Zeng et al., 2022) is a VLM trained to align visual-text data in a more localized way. In comparison with other models, it learns from four types of data: box-object and noun labels, region and text annotations, image-caption data, and video-text data. $X^2$-VLM consists of vision, text, and a multimodal cross-attention module. The vision encoder is initialized with BEiT-2 Peng et al. (2022) while the text encoder is initialized with BERT Devlin et al. (2018). $X^2$-VLM is trained using the three objectives: ITC loss, ITM loss, and masked language modeling (MLM) loss.

Table 8: Model details. In the prompt, "{}" is replaced by the question, QA: "Question: {} Short answer:", *: LLaVA prompt (see end of Supp. Sec. A.1), Inference generation hyperparameters: Maximum number of generated tokens and number of beams in beam search. We *mark frozen weights* for methods where the authors only train the Fusion Module.

| Method | Image Res. (px) | Prompt | Visual Encoder | Text Encoder | Fusion Module | Model Size | Max Tokens | Beams |
|---|---|---|---|---|---|---|---|---|
| BLIP$_{vqa}$ | 480 | {} | ViT-B/16 (ImageNet) | BERT$_{base}$ | - | 0.361B | 20 | 5 |
| $X^2$-VLM$_{vqa}$ B | 768 | {} | 12L ViT (BeiT-2) | BERT$_{base}$ | 6 L Trans. | 0.334B | 20 | 1 |
| $X^2$-VLM$_{vqa}$ L | 768 | {} | 24L ViT (BeiT-2) | BERT$_{base}$ | 6 L Trans. | 0.724B | 20 | 1 |
| BLIP-2 OPT | 224 | QA | *ViT-g/1 EVA-CLIP* | *unsup. OPT* | 12L Trans. | 3.745B | 30 | 5 |
| BLIP-2 T5 | 224 | QA | *ViT-g/1 EVA-CLIP* | *inst. FlanT5 XL* | 12L Trans. | 3.942B | 30 | 5 |
| InstructBLIP T5 | 224 | QA | *ViT-g/1 EVA-CLIP* | *inst. FlanT5 XL* | 12L Trans. | 4.023B | 256 | 1 |
| InstructBLIP V | 224 | QA | *ViT-g/1 EVA-CLIP* | *Vicuna-7B* | 12L Trans. | 7.913B | 256 | 1 |
| LLaVA | 224 | * | *ViT-L/14 CLIP* | *LLaMA 7B* | 1 Linear L | 6.743B | 1024 | 1 |

**LLaVA prompt.** "You are LLaVA, a large language and vision assistant trained by UW Madison WAIV Lab. You are able to understand the visual content that the user provides, and assist the user with a variety of tasks using natural language. Follow the instructions carefully and explain your answers in detail.
Human: Hi!
Assistant: Hi there! How can I help you today?
Human: {}"

**Handling long model outputs.** The EVA-CLIP model used by the *ClipM* metric has a maximum input length of 77 tokens. Also, the *Cont* metric might become unreliable for very long outputs since it rates a ground truth label as correct if contained in the output, regardless of the output length. Therefore, we reduce all model predictions on all datasets to a maximum of about 45 words. We use the `nltk` (Loper & Bird, 2002) English tokenizer to split model predictions into sentences and try to cut off the prediction at a sentence boundary to a total length of 40–50 words. If there is no sentence boundary in that range, we cut off after 45 words. This only affects the LLaVA model, which tends to produce long paragraphs of answers. On ImageNet-oVQA output, this procedure reduces the average length of LLaVA outputs from 76 to 44 and affects 30% of the input.

**Cut-off points.** We observe that BLIP-2 OPT often generates outputs in the form of "(answer) Long answer: (reformulated answer)" after being prompted with "Question: (question) Short answer: ". We consider this a signal that the model has finished answering and only evaluate all text that appears before the model generates "Long answer: " or "Short answer: ". On ImageNet-oVQA output, this procedure affects 25% of the input.

**Follow-up.** After determining the candidate parent class for the follow-up, we calculate the similarity between the model prediction and all parent synonyms using EVA-CLIP. Since our goal is to only ask follow-up questions about concepts that the model already described, we only ask a follow-up question if the most similar synonym has a similarity of at least $\delta$. We set the hyperparameter $\delta = 0.37$ based on qualitative evaluation. For ImageNet, we create the parent embeddings using prompt templates described in Supp. Sec. A.2. For ActivityNet, we do not use prompt templates.

## A.2 DATASET-SPECIFIC DETAILS

We report statistics over all evaluated datasets in Supp. Tab. 9. We also provide an overview of all questions we ask in Supp. Tab. 10 and Supp. Fig. 3.

**COCO-oVQA.** COCO contains bounding boxes for each object instance among the 80 object categories. We use the "validation 2017" set to build the COCO-oVQA task, which consists of 36,781 object instances. For each instance, we crop the object using the ground-truth bounding box considering a minimum side of 40 pixels leaving a margin of 4 pixels on each side. We use three different questions to ask the VLMs about the object category: "What can be seen in the image?", "What is in the image?" and "What's this?". For the *ClipMatch* metric, we obtain a single class embedding by averaging the embeddings of 80 different prompt templates such as `A photo of a {label}` and `A good photo of the {label}` as done by Radford et al. (2021).

**ImageNet-oVQA.** ImageNet provides bounding boxes for all objects corresponding to the ground truth class label of an image. For each image, we select the box with the biggest area and increase it to a minimum size of 64 pixels on each side. Since all models we test require squared inputs, we convert the rectangle boxes to squares by increasing the size of the smaller side. We reuse the same questions as for COCO and the same prompt templates to obtain the embeddings. To build the label hierarchy from WordNet, we only consider parents with more than one child to simplify the hierarchy. We also exclude the root node "entity".

**ActivityNet-oVQA.** We do not use any prompt templates for creating the class embeddings. For asking the follow-up question, we use the label hierarchy provided by the dataset authors and remove the root node "activity".

## A.3 CODE

We build our codebase on top of LAVIS (Li et al., 2023a).

**cleanliness**
How would you describe the object's appearance in terms of cleanliness?
How would you describe the object in terms of cleanliness?
What is the state of the object in terms of cleanliness?
**clothes color**
What colors are the object's clothes in the image?
What colors are the clothes the object is wearing?
What colors do you notice in their attire?
**clothes pattern**
What patterns are the clothes the object is wearing?
What type of pattern is on the object's attire?
How would you describe the design on the object's clothing?
**color quantity**
The number of colors the object has is
How many colors does the object have?
How many distinct colors do you notice on the object?
**color**
What colors are present in the object?
Describe the color of the object.
Identify the color of the object.
**cooked**
What's the condition of the object's cooking process?
How would you describe the condition of the object in terms of cooking?
How cooked is the object?
**face expression**
Which is the face expression of the object?
What is the face expression of the object?
What is the object's facial expression?
**gender**
Describe the gender of the object.
What is the gender of the object?
Which gender is the object?
**group**
How many objects are present in the image?
How many objects are visible?
Are we looking at an individual object or a group?
**hair color**
Which is the hair color of the object?
What is the hair color of the object?
Describe the color of the object's hair.
**hair length**
What is the hair length of the object?
How would you classify the hair length of the object?
Describe the hair length of the object.
**hair tone**
How would you describe the brightness of the object's hair?
Can you specify the lightness or darkness of the object's hair?
What intensity is the object's hair color?

**hair type**
How would you describe the object's hair texture?
What kind of hair texture does the object have?
What is the nature of the object's hair?
**length**
Describe the object's length.
What is the length of the object?
Describe the length of the object.
**material**
What material is the object made of?
What is the material of the object?
Describe the material of the object.
**maturity**
What is the object's maturity level?
What age category does the object belong to?
Which is the maturity of the object?
**optical property**
How would you describe the object's ability to transmit light?
What is the optical property of the object?
What is the object's optical property?
**order**
Does the object exhibit a systematic arrangement or a more disordered appearance?
What is the state of organization of the object?
How would you describe the arrangement of the object?
**pattern**
Which is the pattern of the object?
Which pattern does the object have?
What pattern is the object?
**position**
What is the position of the object?
Describe the position of the object.
What position is the object in?
**size**
How would you describe the size of the object?
Describe the object's size.
What size is the object?
**state**
Which is the state of the object?
What is the state of the object?
Describe the state of the object.
**texture**
Describe the texture of the object.
What is the feel of the object's surface?
Describe the object's texture.
**tone**
How would you describe the object's color intensity?
What is the lightness or darkness of the object?
Describe the tone of the object.

Figure 3: **OVAD-oVQA questions.** For every attribute type we build three different questions to evaluate the Text-VLMs. We show an example of the three different questions for every type of attribute. The word object is replaced by the noun category for every question.

Table 9: **Dataset details.** We report details for the *balanced-testdev* set on GQA and for the *val* set on all other datasets. For ActivityNet we report the number of videos and segments instead of images and crops. During evaluation on ActivityNet we consider the center frame of a segment as input for the model. For OVAD we evaluate 123k attribute annotations with 3 questions each. *Q*: question, *A*: answer, *len*: length (number of words), *Cls*: classes.

| Dataset | Imgs (k) | Crops (k) | Q (k) | Q len | ± | Uniq Q | A (k) | A len | ± | Uniq A | Cls |
|---|---|---|---|---|---|---|---|---|---|---|---|
| ImageNet | 50.0 | 50.0 | 150.0 | 5.00 | 2.00 | 3 | 150.0 | 1.67 | 0.74 | 1000 | 1000 |
| COCO | 5.0 | 36.8 | 110.3 | 5.00 | 1.63 | 3 | 110.3 | 1.10 | 0.30 | 80 | 80 |
| ActivityNet | 4.9 | 7.7 | 23.0 | 4.33 | 1.25 | 3 | 23.0 | 2.02 | 0.84 | 200 | 200 |
| OVAD | 2.0 | 14.3 | 369.0 | 7.61 | 1.77 | 3099 | 1229.9 | 1.08 | 0.27 | 249 | 117 |
| VQAv2 | 40.5 | 214.4 | 214.4 | 6.23 | 1.96 | 81272 | 2143.5 | 1.17 | 0.57 | 90805 | - |
| GQA | 0.4 | 12.6 | 12.6 | 8.53 | 3.23 | 11484 | 12.6 | 1.06 | 0.24 | 722 | - |

Table 10: **Overview of questions.** When asking follow-up questions, we replace the {placeholder} with the corresponding object or activity to ask about e.g. "What type of *dog* is this?"

| Dataset | Question | Follow-up question |
|---|---|---|
| ImageNet | What can be seen in the image? What is in the image? What's this? | What type of {object} is this? |
| COCO | What can be seen in the image? What is in the image? What's this? | n/a |
| ActivityNet | What activity is this? What is happening in the image? What is this? | What type of {activity} is this? |
| OVAD | See Supp. Fig. 3 | n/a |

## B  ADDITIONAL RESULTS

We report all results on ImageNet without cropping using retrieval models in Supp. Tab. 11 and using VQA models in Supp. Tab. 12.

We compare the hyperparameters of BLIP-2 both in the default and VQA settings. The default setting is shown in Supp. Tab. 8. For VQA, Li et al. (2023b) apply an exponential length penalty $\lambda = -1$ that promotes generating shorter answers and limits the maximum number of tokens to 10 in order to match better the *EM* metric used in VQAv2 and GQA. Supp. Tab. 13 shows that the hyperparameters optimized for VQA benchmarks produce significantly shorter answers and worse results on our ImageNet-oVQA benchmark.

### B.1  QUALITATIVE EXAMPLES

We show qualitative examples of our oVQA benchmark for ImageNet in Supp. Fig. 4, ActivityNet in Supp. Fig. 5, OVAD in Supp. Fig. 6, and for COCO in Supp. Fig. 7. Additionally, we show classical VQA examples evaluated with the different metrics in Supp. Fig. 8. Finally, we show a qualitative overview of all evaluated datasets in Supp. Fig. 9.

Table 11: Zero-shot retrieval results for ImageNet (not cropped).

| Model | R@1 | R@5 |
|---|---|---|
| CLIP ViT-L-14 | 75.52 | 94.57 |
| CLIP EVA01-g-14 | 78.52 | 95.50 |
| BLIP-2 | 63.59 | 87.72 |
| BLIP-2 (COCO ft) | 62.26 | 85.31 |

Table 12: ImageNet (not cropped) classification accuracy using VQA models. **Top:** Results for asking the questions "What can be seen in the image?", "What is in the image?" and "What's this?" **Bottom:** Results for asking "What type of *object* is this?" as a response to the output of the first question. We highlight the **first**, second and *third* best method.

| | Method | ClipM@1 | ± | ClipM@5 | ± | Cont | ± | EM | ± |
|---|---|---|---|---|---|---|---|---|---|
| 1st Question | BLIP (vqa finetuned) | 23.71 | 0.88 | 42.94 | 1.66 | 9.40 | 0.90 | *8.27* | 1.37 |
| | $X^2$-VLM$_{vqa}$ B | 20.30 | 1.86 | 39.47 | 3.00 | 7.60 | 1.37 | 7.10 | 1.48 |
| | $X^2$-VLM$_{vqa}$ L | 22.77 | 2.59 | 42.31 | 3.93 | 8.71 | 1.70 | 7.97 | 1.94 |
| | BLIP-2 OPT | **49.02** | 1.43 | 69.02 | 1.04 | **27.89** | 1.17 | 0.83 | 0.20 |
| | BLIP-2 T5 | 45.53 | 1.42 | *65.37* | 1.44 | 23.66 | 1.76 | 1.55 | 1.38 |
| | InstructBLIP T5 | 33.48 | 4.07 | 52.78 | 4.78 | 15.24 | 2.78 | **14.64** | 2.79 |
| | InstructBLIP V | 32.03 | 2.48 | 51.51 | 2.92 | 14.77 | 2.59 | 14.14 | 2.54 |
| | LLaVA | *42.84* | 0.58 | **69.91** | 0.93 | *21.24* | 0.54 | 0.00 | 0.00 |
| 1st Question | BLIP (vqa finetuned) | 34.44 | 0.22 | 53.02 | 0.20 | 14.48 | 0.60 | *13.07* | 1.08 |
| | $X^2$-VLM$_{vqa}$ B | 29.03 | 0.29 | 47.76 | 0.21 | 11.27 | 0.53 | 10.59 | 0.66 |
| | $X^2$-VLM$_{vqa}$ L | 33.42 | 0.46 | 52.57 | 0.29 | 13.43 | 0.70 | 12.47 | 0.93 |
| | BLIP-2 OPT | **60.59** | 1.29 | **78.22** | 0.97 | **34.02** | 0.21 | 2.83 | 0.54 |
| | BLIP-2 T5 | 58.69 | 0.55 | *75.89* | 0.33 | 29.71 | 1.05 | 5.58 | 0.88 |
| | InstructBLIP T5 | 49.53 | 0.98 | 67.60 | 0.58 | 23.18 | 1.04 | **22.41** | 1.08 |
| | InstructBLIP V | 48.20 | 0.89 | 66.03 | 0.60 | 22.30 | 1.72 | 21.47 | 1.69 |
| | LLaVA | *54.92* | 0.21 | 77.67 | 0.12 | *25.51* | 0.22 | 0.00 | 0.00 |

Table 13: ImageNet (cropped) classification comparing different hyperparameter setups for BLIP-2.

| | Method | ClipM@1 | ± | ClipM@5 | ± | Cont | ± | EM | ± | Len | ± |
|---|---|---|---|---|---|---|---|---|---|---|---|
| 1st Q. | OPT$_{vqa}$ | 35.32 | 13.99 | 50.31 | 18.92 | 18.67 | 8.10 | 6.67 | 1.76 | 1.96 | 0.37 |
| | OPT | **57.10** | 1.70 | **77.24** | 1.66 | **35.49** | 1.67 | 0.87 | 0.31 | 7.94 | 0.94 |
| | T-5$_{vqa}$ | 43.02 | 2.51 | 62.75 | 2.51 | 21.76 | 2.22 | *13.96* | 4.09 | 1.56 | 0.24 |
| | T-5 | 54.50 | 1.92 | 75.71 | 1.85 | 30.05 | 2.36 | 1.77 | 1.30 | 4.21 | 0.52 |
| Foll. Q. | OPT$_{vqa}$ | 50.93 | 8.77 | 66.26 | 9.96 | 26.95 | 5.05 | 12.01 | 1.09 | 1.94 | 0.07 |
| | OPT | **67.22** | 0.27 | **83.54** | 0.12 | **40.31** | 0.74 | 2.54 | 0.22 | 6.53 | 0.38 |
| | T-5$_{vqa}$ | 55.51 | 1.32 | 72.26 | 1.05 | 27.84 | 1.01 | *19.96* | 3.81 | 1.32 | 0.08 |
| | T-5 | 64.71 | 0.35 | 80.83 | 0.16 | 34.43 | 1.22 | 4.81 | 0.51 | 3.17 | 0.20 |

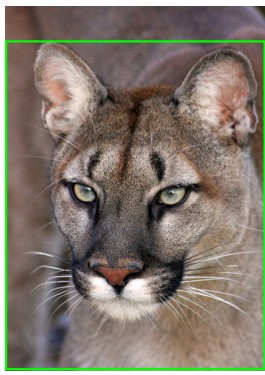

*Question:* What's this?
*Label:* cougar
**BLIP-2 T5** *output*: a mountain lion

**ClipM Top-5 similarities**:
cougar (*0.792*)
lion (*0.682*)
snow leopard (*0.636*)
lynx (*0.527*)
leopard (*0.511*)

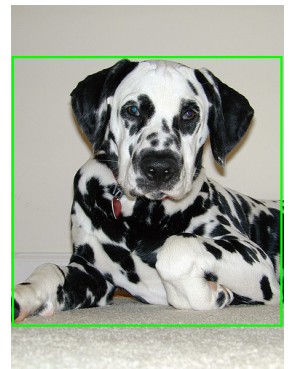

*Question:* What's this?
*Label:* Dalmatian
**BLIP-2 OPT** *output*: it's a dalmatian
**LLaVA** *output*: The image features a large black and white dog laying down on the floor, possibly on a carpet.
*Follow-up Question:* What type of dog is this?
**LLaVA** *output*: The dog in the image is a Dalmatian.

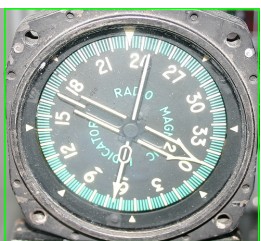

*Question:* What's this?
*Label:* magnetic compass
**BLIP-2 T5** *output*: a radio compass

**ClipM Top-5 similarities**:
magnetic compass (*0.781*)
radio (*0.701*)
barometer (*0.629*)
odometer (*0.517*)
stopwatch (*0.513*)

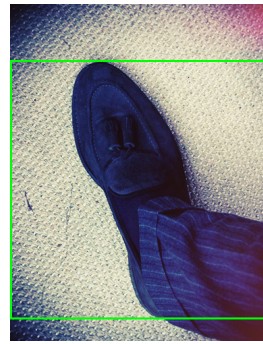

*Question:* What's this?
*Label:* slip-on shoe
**X$^2$-VLM$_{vqa}$ L** *output*: shoe
*Follow-up Question:* What type of shoe is this?
**X$^2$-VLM$_{vqa}$ L** *follow-up output*: black

(a) *ClipM* allows evaluating predictions as correct for which text-matching metrics like *EM* and *Cont* fail.

(b) **Top:** Follow-up questions improve the fairness of the evaluation, since they reduce the chance of punishing reasonable answers. **Bottom:** If a model cannot produce the required level of detail, it will still fail the evaluation.

Figure 4: Qualitative results for ImageNet-oVQA. Only the bounding box crop is considered as input image for the model prediction. Coloring: Answers are considered correct / wrong under *ClipM* metric.

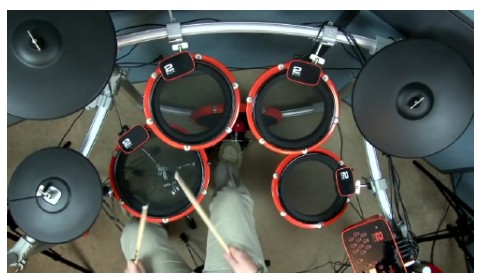

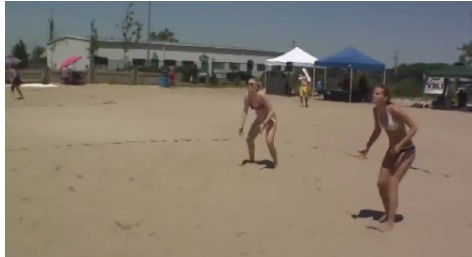

*Question:* What activity is this?
*Label:* playing drums
**BLIP_{vqa}** *output*: playing music
*Hierarchy:* playing musical instruments
*Follow-up Question:* What type of *playing musical instruments* is this?
**BLIP_{vqa}** *follow-up output*: drums

(a)

*Question:* What activity is this?
*Label:* beach volleyball
**BLIP-2 OPT** *output*: volleyball
*Hierarchy:* playing volleyball
*Follow-up Question:* What type of *playing volleyball* is this?
**BLIP-2 OPT** *follow-up output*: beach volleyball

(b)

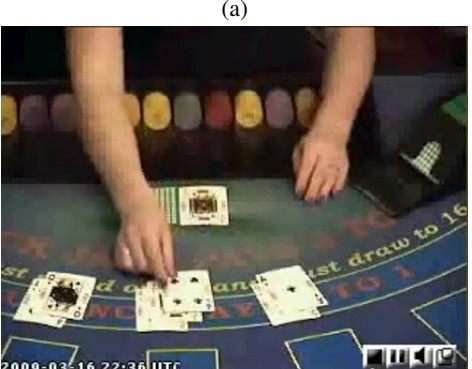

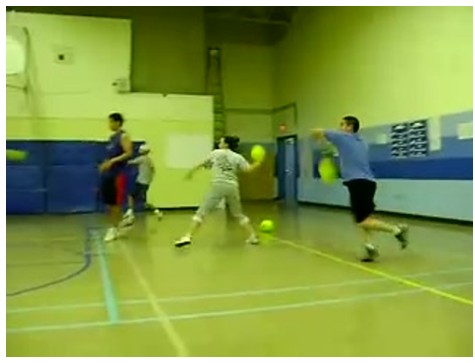

*Question:* What is this?
*Label:* playing blackjack
**BLIP_{vqa}** *output*: business
**BLIP-2 OPT** *output*: it's a blackjack
**InstructBLIP T5** *output*: blackjack
**LLaVA** *output*: The image shows a person sitting at a table, shuffling a deck of cards...
**X$^2$-VLM_{vqa} L** *output*: table

(c)

*Question:* What is this?
*Label:* dodgeball
**BLIP_{vqa}** *output*: tennis
**BLIP-2 OPT** *output*: dodgeball
**InstructBLIP T5** *output*: frisbee
**LLaVA** *output*: This is a group activity taking place in a gym where people are playing with basketballs and volleyballs.
**X$^2$-VLM_{vqa} L** *output*: soccer

(d)

Figure 5: Qualitative examples for the Activity oVQA task. Figures (a) & (b) show how the follow-up question makes use of the hierarchical parent to find the exact answer. Figures (c) & (d) show the outputs of different VLM models for the Activity-oVQA task. Coloring: Answers are considered correct / wrong under *ClipM* metric.

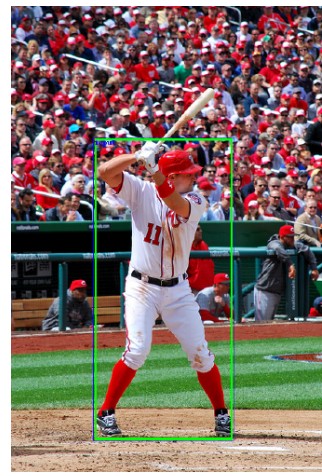

*Question:* What is the position of the person?
*Label:* standing / upright / vertical
**BLIP_vqa** *output*: standing
**BLIP-2 OPT** *output*: 1st base
**InstructBLIP T5** *output*: batter
**LLaVA** *output*: The person is in the position of a batter, standing at the home plate ...
**X$^2$-VLM_vqa L** *output*: standing

(a)

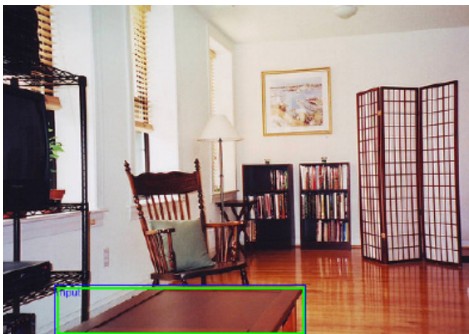

*Question:* Which texture does the dining table have?
*Label:* sleek / smooth
**BLIP_vqa** *output*: smooth
**BLIP-2 OPT** *output*: it feels like wood, but it's not wood, it's laminate
**InstructBLIP T5** *output*: wood
**LLaVA** *output*: The dining table's surface is made of wood, giving it a warm and natural appearance.
**X$^2$-VLM_vqa L** *output*: smooth

(b)

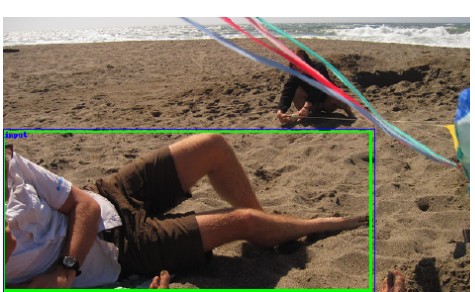

*Question:* How many people are present in the image?
*Label:* individual / one / single / 1 / sole / alone
**BLIP_vqa** *output*: one
**BLIP-2 OPT** *output*: None.
**InstructBLIP T5** *output*: 2
**LLaVA** *output*: There are two people present in the image.
**X$^2$-VLM_vqa L** *output*: one

(c)

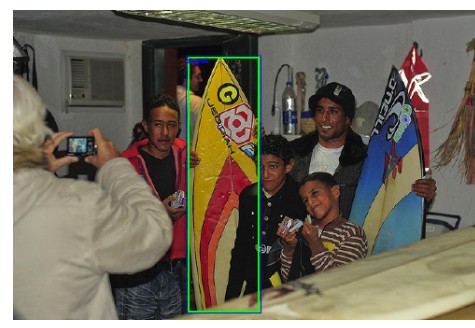

*Question:* How many surfboards are present in the image?
*Label:* individual / one / single / 1 / sole / alone
**BLIP_vqa** *output*: 1
**BLIP-2 OPT** *output*: One.
**InstructBLIP T5** *output*: 1
**LLaVA** *output*: There are two surfboards in the image.
**X$^2$-VLM_vqa L** *output*: 2

(d)

Figure 6: Qualitative examples for the Attribute-oVQA task. BLIP-2 OPT tends to output nouns/objects for many attribute-type questions (see (a) & (b)). We observe that LLaVA tends to hallucinate long answers ignoring the image for contextual questions like quantity of objects in the image (see (c) & (d)). Only the bounding box crop is considered as input image for the model prediction. Coloring: Answers are considered correct / wrong under *Cont Syn* metric.

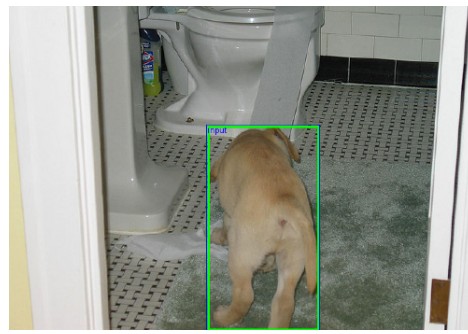

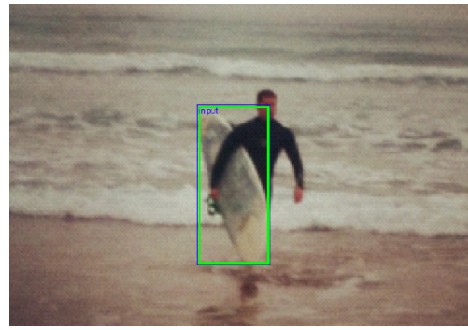

*Question:* What is in the image?
*Label:* dog
**BLIP_vqa** *output*: puppy
**BLIP-2 OPT** *output*: A puppy playing with a toy in the bathroom
**InstructBLIP T5** *output*: dog
**LLaVA** *output*: The image features a large tan dog sitting on the floor, with its back facing the camera (...)
$X^2$-**VLM_vqa L** *output*: puppy

(a)

*Question:* What's this?
*Label:* surfboard
**BLIP_vqa** *output*: dog
**BLIP-2 T5** *output*: a photo of a man holding a surfboard
**InstructBLIP T5** *output*: surfboard
**LLaVA** *output*: The image is a blurry photo of a person holding a surfboard (...) The surfboard is white and extends across the image. (...)
$X^2$-**VLM_vqa L** *output*: man

(b)

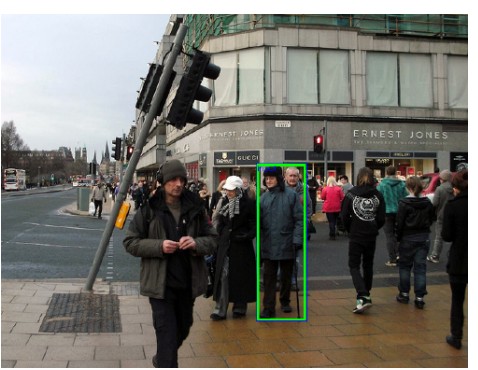

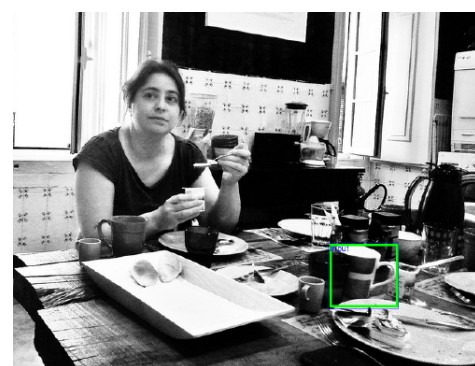

*Question:* What can be seen in the image?
*Label:* person
**BLIP_vqa** *output*: group of people
**BLIP-2 OPT** *output*: a man, a woman, a dog, a cat, a dog, a cat, a cat, a cat, a cat, a cat
**InstructBLIP T5** *output*: a group of people walking down a street
**LLaVA** *output*: In the image, a person is walking down the street (...) The person is wearing a heavy winter coat, gloves, and a hat.
$X^2$-**VLM_vqa L** *output*: people walking

(c)

*Question:* What can be seen in the image?
*Label:* cup
**BLIP_vqa** *output*: coffee cup and saucer
**BLIP-2 OPT** *output*: a plate, a cup, a knife, a fork, a spoon, a knife, a fork ...
**InstructBLIP T5** *output*: coffee cup
**LLaVA** *output*: In the image, there are two coffee cups sitting on a table, with one of them being a white coffee mug with a heart shape on it.
$X^2$-**VLM_vqa L** *output*: mug

(d)

Figure 7: Qualitative examples for the COCO oVQA task. We observe that ClipMatch can consider synonyms and longer texts as positive matches (see (a) for puppy & (b) for container). In (c) and (d), we observe how BLIP-2 OPT tends to repeat many nouns instead of simply the correct one and how LLaVA generates detailed answers but sometimes hallucinates. Only the bounding box crop is considered as input image for the model prediction. Coloring: Answers are considered correct / wrong under *Cont Syn* metric.

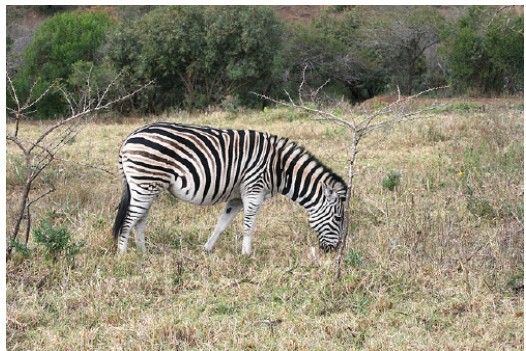

*Question:* What is the zebra eating?
*Label:* grass (x9), dry grass (x1)
**BLIP-2 T5** *output*: grass
**LLaVA** *output*: The zebra is eating grass in the field.

| Output | Label | EM | Cont | LLaMA-2 | GPT-4 |
|---|---|---|---|---|---|
| grass | grass | 1.00 | 1.00 | 1.00 | 1.00 |
| The zebra is eating grass in the field. | grass | 0.00 | 1.00 | 1.00 | 1.00 |
| grass | dry grass | 0.00 | 0.00 | 0.75 | 0.75 |
| The zebra is eating grass in the field. | dry grass | 0.00 | 0.00 | 0.75 | 0.75 |

**VQAv2** with 10 human answers. Metrics are evaluated for each answer and then aggregated as described in Sec. 3.2. **Row 1:** All metrics give full points if the model output matches exactly. **Row 2:** *EM* cannot properly judge the output of LLaVA while *Cont* and both LLM-based metrics successfully evaluate the output as correct. **Rows 3, 4:** Both text-based metrics cannot judge either output as correct, since the label is longer than the prediction and therefore can neither match nor be contained in the prediction. The LLM-based metrics still succeed in judging the output.

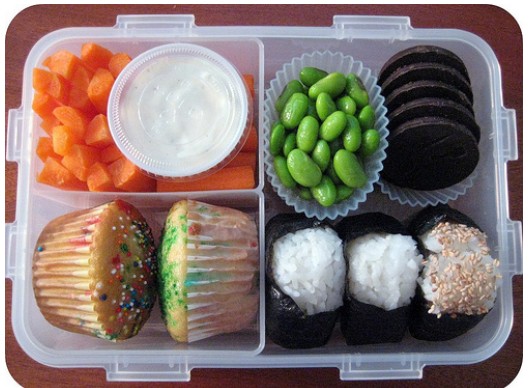

*Question:* What are the vegetables to the left of the bowl that is to the left of the cookies?
*Label:* carrots
**BLIP-2 T5** *output*: carrots
**LLaVA** *output*: The vegetables to the left of the bowl are carrots and green beans.

| Output | Label | EM | Cont | LLaMA-2 | GPT-4 |
|---|---|---|---|---|---|
| carrots | carrots | 1.00 | 1.00 | 1.00 | 1.00 |
| The vegetables to the left of the bowl are carrots and green beans. | carrots | 0.00 | 1.00 | 1.00 | 0.25 |

**GQA** with 1 label. **Row 6:** *Cont* and LLaMA-2 overestimate the performance of LLaVA and do not penalize the incorrect "green beans" part of the answer, while GPT-4 penalizes it.

Figure 8: Qualitative examples for different metrics. Metric scores range from 0.00 (worst) to 1.00 (best).

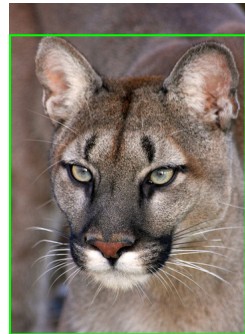

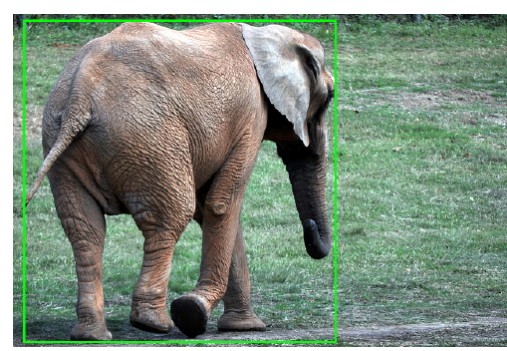

*Dataset:* ImageNet
*Question:* What's this?
*Label:* cougar

*Dataset:* COCO
*Question:* What's this?
*Label:* elephant

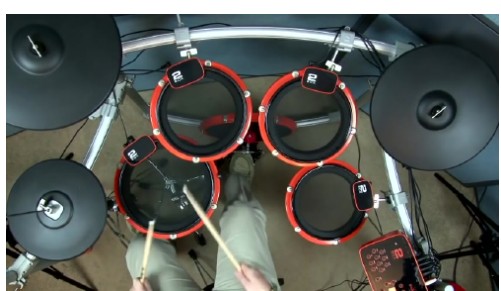

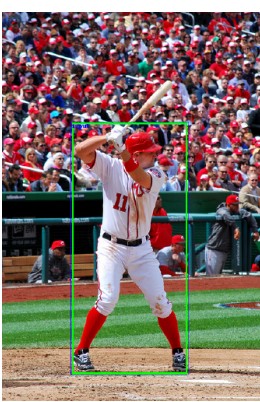

*Dataset:* ActivityNet
*Question:* What activity is this?
*Label:* playing drums

*Dataset:* OVAD
*Question:* What is the position of the person?
*Label:* standing / upright / vertical

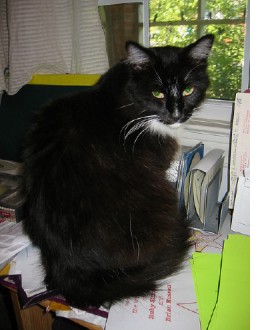

*Dataset:* VQAv2
*Question:* Where is the cat?
*Label:* on desk (x4), desk (x3), center of picture, at home, on table

*Dataset:* GQA
*Question:* What is the spoon made of?
*Label:* metal

Figure 9: Qualitative comparison between all datasets used in the benchmark.

## C  HUMAN EVALUATION

We base our human study design on Chen et al. (2020) and ask humans to rate model predictions from 1 (completely wrong) to 5 (completely correct).

**Data selection.** We use the VQAv2 validation dataset for our human evaluation and select questions from each of the three subsets: 50% "other", 25% "number" and 25% "yes/no". To avoid very noisy questions, we analyze the 10 human answers per question and only select questions with at least 4 of 10 humans agreeing for "other" and "number" questions and at least 9 of 10 humans agreeing for "yes/no" questions. We keep the top answer for each question as the ground truth answer. In total, we select 1000 questions.

**Choice of model predictions.** A good text semantic metric should work well on both short and long answers. Therefore, we select the two models BLIP-2 T5 with 1.6 words on average per answer and LLaVA with 12.6 words on average per answer for our study.

**Processing.** For metrics based on text heuristics (*Cont*, *EM* and last 5 rows of Tab. 7), we preprocess the model predictions as the authors of the dataset VQAv2, described in Sec. 3.2. To reduce the human workload, we automatically label all predictions that are correct under *EM* as "5" and all empty model outputs as "1". When giving predictions to humans or deep learning-based models, we do not preprocess the outputs since we hypothesize that those judges can use certain punctuation or model-specific cues, e.g. preserving upper case letters. Given a score $s$ in the range from 1 to 5, we convert it into the range from 0 to 1 as $s_{\text{norm}} = (s - 1)/4$.

**Instructional examples.** To create examples of the task for human workers and GPT-4, we sample a second set of data in the same way. We manually judged 100 examples on the second set and commented on our reason for the rating on 20 of them.

**Human evaluation.** We prompt the human workers with the instructions shown in Supp. Fig. 10 and present them the 20 commented examples, which contain the fields "Question", "Correct answer", "Predicted answer", "Correct rating" and "Reason". We run the evaluation internally. Each datapoint is annotated by 3 workers.

**Majority voting and inter-annotator agreement.** Same as Chen et al. (2020) we calculate inter-annotator agreement using Krippendorff's Alpha-Reliability (Hayes & Krippendorff, 2007) and obtain 0.93, indicating strong agreement. We use majority voting to receive the final gold standard annotation as done by Bai et al. (2023). For the 4.6% of cases where all 3 annotators disagree, we average and round the votes.

**GPT-4 evaluation.** We use the manually judged 100 examples as in-context few-shot prompts for GPT-4. To construct the input prompt, we experimentally find an upper limit on the input length where GPT-4 starts to produce inconsistent outputs and find the output to be consistent for 10 few-shot examples and 30 datapoints in one prompt. An output is considered consistent if it matches the exact output schema defined in the prompt in Supp. Fig. 11. Inputting new datapoints in a dialogue style without repeating the instructions resulted in GPT-4 forgetting the task and producing inconsistent outputs, so we simply restart the dialogue for each new group of 30 datapoints. At each iteration, we randomly sample 30 datapoints and 10 few-shot examples and input them into GPT-4 as described until all datapoints are annotated. In the 0-shot experiment, we remove the "Example input:" and "Example output:" sections from the input prompt.

**GPT-4 monetary evaluation cost.** To approximate a lower bound on the cost of such an evaluation, we calculate the average number of words in the input and output. We ignore the JSON formatting characters since we assume a cost-optimal evaluation would choose a format with less such characters. OpenAI (2023b) reports 1K tokens as representing about 750 words and sets the price of their API at $0.03 / $0.06 per 1K input / output tokens, respectively. On average, we have 1720 / 970 input / output tokens for a batch of 30 predictions, resulting in a cost of about $3.66 for 1000 predictions.

**LLaMA-2 evaluation.** For each model output to rate, we randomly select 5 in-context few-shot examples from the manually judged 100 examples and format them as described in Supp. Fig. 12. We use the 70B version and generate one single token with greedy sampling. If the token is not an integer, we set the score to 1. If the token is an integer above 5 or below 1, we set the score to 5 or

---

1. Read the question, the correct answer and the predicted answer.
2. Select the score that best reflects how closely the predicted answer captures the same information as the correct answer.

Note: Please try to not think about what the image could have looked like. This is a text-only task. All you know about the image is the content of the reference.
In the rare cases where the question does not make sense, simply compare the predicted answer to the correct answer and ignore the question.

Scores:

1: completely wrong
2: mostly wrong
3: half right
4: mostly right
5: completely right

---

Figure 10: Instructions for the human metric annotation task on the VQAv2 dataset.

1 respectively. However, in $\geq$ 99% of cases, LLaMA-2 produces an integer between 1 and 5. In the 0-shot experiment, we remove the 5 in-context examples from the input prompt.

**Results.** *GPT-4* performs up to par with individual workers in both the short and long prediction case. However, it depends on an external proprietary service and with $3.66 for 1000 predictions, evaluation quickly becomes costly when evaluating several models and datasets. LLaMA-2 is a good option that does not depend on an external service but is still computationally expensive with 70B parameters. Notably, LLaMA-2 requires in-context examples to perform well. *EM* as the de facto standard metric for VQA performs reasonably well in the short answer case. However, *EM* completely fails in evaluating the long answers from LLaVA. We find *Cont* to be the best choice for a metric that can be evaluated inexpensively when evaluating short ground truth answers and mixed length model answers. However, it may overestimate the performance of models that list several guesses. We test the extreme case and find that listing the top 20 training set answers for every question achieves 59.91% *Cont* on VQAv2 (val set).

We also consider *n-gram based* metrics that are used traditionally for evaluating translation models: BLEU, CIDEr, METEOR, and ROUGE. These metrics match n-tuples of words between model prediction and true answer. However, potentially due to the ground truth answers in VQAv2 being very short, they do not improve over the more simple *EM* and *Cont*. We also observe interesting cases, as in METEOR, where the correlation to human judgment is rather high when evaluating only short or long answers but drops when evaluating mixed lengths of answers. This behavior indicates a bias to the answer length which only diminishes the correlation once the answer length varies.

Finally, we compare with 3 *learned* metrics. BertScore (Zhang et al., 2020) computes token similarity between embeddings created with RoBERTa (Liu et al., 2019) while BLEURT-20 (Pu et al., 2021) and LERC (Chen et al., 2020) directly learn to output human judgment scores via regression. However, none of these models beat the *Cont* baseline. In conclusion, depending on the budget and size of the evaluation, GPT-4 and LLaMA-2 are strong LLM-based choices. For more efficient evaluation, *Cont* is a more dependable option compared to *EM* however it is potentially vulnerable against hyperparameter optimization.

---

**User**: You are an annotator that judges the output of a QA system. Instructions:

*(instructions for the dataset)*

Please output JSON format and do not output anything else.

Example input:

```
[{"question": "Is the skateboard in the air?", "correct_answer": "no",
"predicted_answer": "Yes, the skateboard is in  the air, and the
girl is sitting on the ground with her feet on the skateboard."},
(...)
]
```

Example output:

```
[{(same fields as input), "score": 1},
(...)
]
```

Real input:

*(actual inputs, same format as example inputs)*

Real output:

---

**GPT-4**: *(actual outputs, same format as example outputs)*

---

Figure 11: Input prompt for GPT-4 as evaluator.

---

**User**: You are an annotator that judges the output of a QA system. Instructions:

*(instructions for the dataset)*

```
Question: Is the skateboard in the air?
Candidate: Yes, the skateboard is in  the air, and the girl is sitting on
the ground with her feet on the skateboard.
Reference: no
Vote: 1
```

*(4 more in-context examples)*

```
Question: What type of donut is on the top right?
Candidate: chocolate glazed donut
Reference: chocolate iced glazed
Vote:
```

---

**LLaMA-2**: 4

---

Figure 12: Input prompt for LLaMA-2 as evaluator. In this example, LLaMA-2 rates the candidate as 4 of 5 (i.e. 80% accuracy).

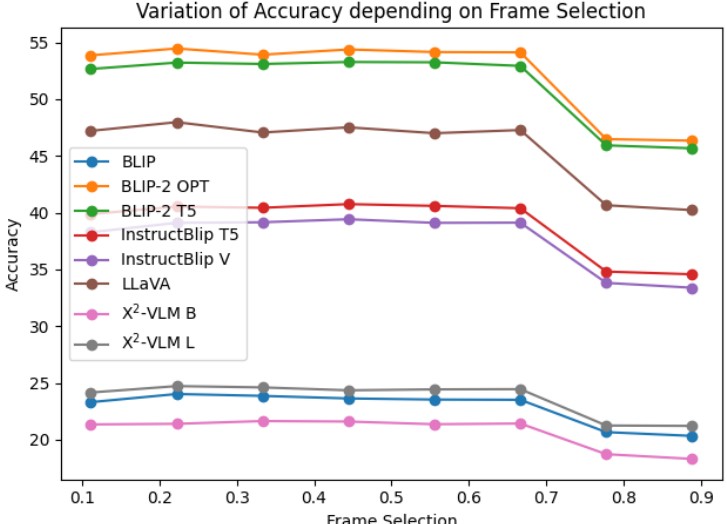

Figure 13: Accuracy of models on ActivityNet-oVQA dataset when varying the frame selection.

## D  VIDEO FRAME SELECTION

To determine the effectiveness of selecting the middle frame among all others, we conduct an ablation study, testing all models over nine frame selections by dividing the segment into nine parts and selecting the closest frame. The results are shown in Supp. Fig. 13. We measure accuracy as the average response to the same three initial questions across all models. The results suggest that frame selection does not significantly impact performance, especially since frames are selected from short segments in which the relevant action occurs.

## E  LIMITATIONS

Our open-ended VQA evaluation still provides room for improvement. Supp. Fig. 14 shows examples of failure cases. Supp. Fig. 14(a) shows the weakness of the Contains metric when listing all possible answers, while Supp. Fig. 7(c and d) show that the ClipM metric can deal with such scenarios by marking the answers as wrong. Supp. Fig. 14(b) shows how the label space of the COCO dataset unfairly penalizes correct answers such as "green beans" for the question "What can be seen in the image?". Supp. Fig. 14(c) shows how ClipM can produce sub-optimal similarities. The model answer describes a red slide that should match the activity "fun sliding down", but ClipM gives higher weight to other playground activities.

Finally, Supp. Fig. 14(d) shows two interesting cases: The output "a fossilized leaf" is associated with animals that often occur as fossils and therefore shows a high similarity to the marine animal "trilobite". On the contrary, the term "fossil" seems to be confused with the digital watch brand of the same name and therefore produces wrong similarities.

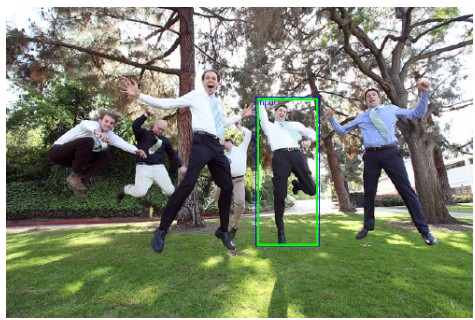

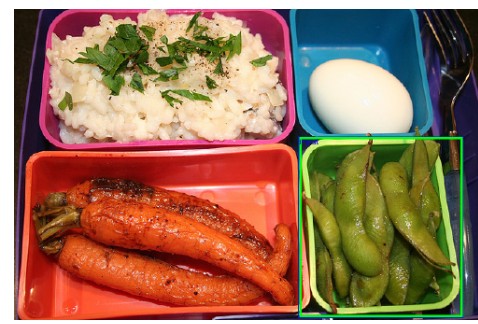

*Dataset:* OVAD-oVQA
*Question:* What colors do you notice in their attire?
*Label:* white
**BLIP-2 OPT** *output*: Black, **white**, gray, brown, green, blue, purple, red, orange, yellow (...)

(a)

*Dataset:* COCO-oVQA
*Question:* What can be seen in the image?
*Label:* bowl
**BLIP$_{vqa}$** *output*: green beans
**LLaVA** *output*: The image shows a tray or **bowl** filled with green beans (...)

(b)

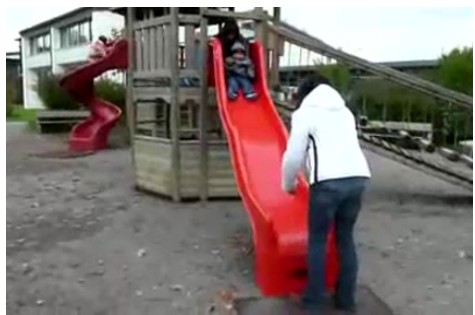

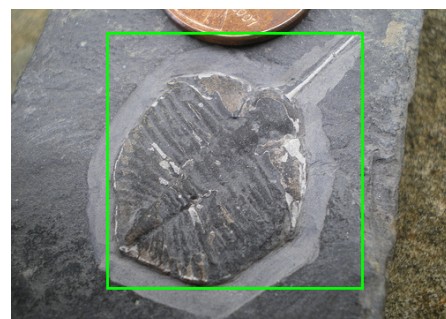

*Dataset:* ActivityNet-oVQA
*Question:* What is happening in the image?
*Label:* Fun sliding down
**LLaVA** *output*: In the image, a woman is helping a child get onto a red **slide** at a park, ensuring the child's safety and guiding them as they climb onto the **playground** equipment.
**ClipM Top-3 similarities**:
Swinging at the playground (*0.495*)
Using the monkey bar (*0.379*)
Fun sliding down (*0.349*)

(c)

*Dataset:* ImageNet-oVQA
*Question:* What can be seen in the image?
*Label:* trilobite
**BLIP-2 T5** *output*: a fossilized leaf
**ClipM Top-3 similarities**:
trilobite (*0.551*)
flatworm (*0.539*)
sea slug (*0.502*)
**InstructBLIP T5** *output*: fossil
**ClipM Top-1 similarities**:
digital watch (*0.322*)
purse (*0.322*)
fig (*0.310*)

(d)

Figure 14: Examples of the limitations of the oVQA benchmark. Only the bounding box crop is considered as input image for the model prediction. Coloring: Answers are considered correct / wrong under *Cont Syn* metric. Refer to the text for the discussion of the examples.

