# OpenReview forum: "Open-ended VQA benchmarking of Vision-Language models by exploiting Classification datasets and their semantic hierarchy"
_ICLR.cc/2024/Conference — ICLR 2024 spotlight_

### Official Review · Reviewer_GTBt · 2023-10-31

**Soundness:** 4 excellent
**Presentation:** 4 excellent
**Contribution:** 4 excellent
**Rating:** 8
**Confidence:** 5

**Summary:**

The work proposes a more principled evaluation of vision-language models by taking advantage of the semantic hierarchy of textual labels. The core problem the work addresses is that the exact match scoring systems used by VQA benchmarks can be ambiguous, and penalize models for essentially correct answers. To fix this, the authors crop images so to the object of interest, allow coarse answers, and ask follow up questions to clarify the fine-grained answer. Finally, they also measure performance with multiple metrics, such as BertScore and GPT-4 based evaluation.

**Strengths:**

While a substantial body of work exists on designing better benchmarks for VQA, the idea of using the semantic hierarchy to ask follow up questions is original. VQA is hard to evaluate, yet is the most fundamental task in vision-language (other than perhaps, image-text matching). One of the biggest problems in VQA is that questions can be underspecified or ambiguous and there can be multiple correct answers to questions.

The method presented here is a neat way to deal with these ambiguities and inconsistencies in the evaluation process.

Another useful contribution of the paper is the empirical evaluation of vision-language models on different aspects of vision-language, such as attribute recognition and coarse vs fine-grained recognition. This is useful.

Finally, Table 7 is also very useful, especially because it helps resolve questions about the appropriateness of different metrics for open-ended VQA scoring.

**Weaknesses:**

There are no substantial weaknesses.

A figure showing the differences between a VQA question + label and examples from the proposed datasets would be useful. Fig. 2 would be easier to read if it said "attribute", "object" etc instead of OVAD, COCO.

**Questions:**

I have no questions.

---

> ### Author Response · Authors · 2023-11-18
>
> Thank you for your constructive feedback on the paper. We appreciate your recognition of our novel approach to VQA and the usefulness of our empirical evaluation. As suggested, we created a figure comparing our proposed datasets with traditional VQA benchmarks to provide a clear overview (Figure 6 in the Supplementary). We also expanded Table 7 to include an in-context metric based on LLaMA-2, which comes close to the effectiveness of GPT-4 in scoring VQA outputs. To evaluate the importance of in-context examples for the LLM metrics, we added zero-shot results to Table 7. We updated the implementation details and results of the LLM metrics in Section 4, Supp. Section C and Supp. Figure 9 to reflect those changes. Additionally, we included metric judgements to our qualitative examples for VQA (Figure 5 in the Supplementary). Finally, we revised Figure 2 and updated the labels as suggested.

---

### Official Review · Reviewer_3kX8 · 2023-10-31

**Soundness:** 3 good
**Presentation:** 3 good
**Contribution:** 3 good
**Rating:** 8
**Confidence:** 5

**Summary:**

Evaluating text-generative vision-language models remains an intricate undertaking due to the inherent limitations of current Visual Question Answering (VQA) benchmarks. Motivated by the need for a more comprehensive assessment mechanism, this paper aims to address the ambiguity and specificity issues rampant in open-ended VQA (oVQA) tasks. These challenges emerge, in part, from the diverse ways natural language can express similar ideas and the constraints of current evaluation metrics that often favor shorter responses. To navigate these obstacles, this research introduces a novel VQA benchmark, enriched with subbenchmarks focusing on objects, actions, and attributes. It leverages a follow-up procedure, drawing from existing classification benchmarks, to generate contextually apt questions and expects models to provide answers with the requisite level of detail. Additionally, the study explores various metrics from the VQA and NLP spheres to find an optimal evaluation criterion, validated via human judgment. This holistic approach paves the way for an intricate analysis of leading vision-language models, unearthing their specific strengths and vulnerabilities.

**Strengths:**

1. Innovative Evaluation Methodologies: The research addresses the inadequacies of existing Visual Question Answering (VQA) benchmarks by proposing a new VQA benchmark. This benchmark, derived from well-established visual classification datasets, facilitates a more detailed evaluation of text-generative vision-language models and their comparative performance against discriminative vision-language models.

2. Addressing Coarse Answer Challenges: Recognizing the challenges in evaluating coarse answers in fine-grained tasks, the research introduces a method using the semantic hierarchy of labels. By doing so, it can automatically generate pertinent follow-up questions about the true category, pushing for more accurate and detailed model responses.

3. Enhanced Ambiguity Management: To better handle the inherent ambiguities in open-ended visual questions, a unique follow-up procedure is proposed. By adapting classification benchmarks for oVQA, the model is first provided with an apt visual context, and then, based on its initial response, a further clarifying question is posed using concept hierarchies. This ensures answers with the desired detail and precision.

4. Comprehensive Metric Evaluation: The research undertakes a rigorous examination of various metrics from both the VQA and NLP domains, emphasizing those that treat paraphrasing and synonyms as valid answers. The eventual metric is grounded in a human evaluation study, ensuring that it aligns closely with human judgments and provides a reliable measure for assessing model performances.

**Weaknesses:**

The paper omits certain statistical details regarding the oVQA dataset, such as the distribution of question/answer lengths, the number of entity class labels, and the formats/types of the questions. Given the dataset's general and expansive nature, there is a concern that, despite the introduction of subbenchmarks, it might introduce new challenges that could affect the quality of evaluations, such as issues related to imbalance.

**Questions:**

Please also refer to the previous section.

---

> ### Author Response · Authors · 2023-11-18
>
> Thank you for your helpful feedback. We are grateful for your acknowledgment of our innovative benchmark
> and rigorous examination of metrics. As proposed, we added tables showing the relevant statistics of all datasets
> (Supp. Table 2) and an overview of questions used in our evaluation (Supp. Table 3, Supp. Figure 1).

---

### Official Review · Reviewer_bBwP · 2023-11-02

**Soundness:** 2 fair
**Presentation:** 3 good
**Contribution:** 2 fair
**Rating:** 5
**Confidence:** 3

**Summary:**

The paper proposes a dataset to granularly evaluate the text-generative vision-language models. The authors base the evaluation benchmark on classification datasets where the labels semantic hierarchy are present. The semantic hierarchy is used as the source of generating the follow-up questions.

**Strengths:**

(1) propose a new topic for evaluating VQA models that assess how well the VQA models can classify or recognize fine-grained objects, activity, and attributes

(2) A intuitive solution to constructing a benchmark that contains the class semantic hierarchy based on classification model.

(3) comprehensive evaluation on lots of open sourced VQA systems.

**Weaknesses:**

(1) I believe assessing how granular a VQA system is in good and necessary, however, when evaluating, the current benchmark put additional constrains on the image space, cropping the image to certain objects and using imagenet which is object centric. These constrains largely limited the scope of VQA that is supposed to work on natural use case, for example, natural image QA (VQA v2, vizwiz, etc).

(2) The Cropping activity operations seems very risky the action needs a few frames to evaluate? Like sit down and stand up?

**Questions:**

Please comment on the weakness part.

---

> ### Author Response · Authors · 2023-11-18
>
> Thank you for your constructive feedback and positive comments about our benchmark.
>
> **Answer to question 1**
>
> We agree that our scope of VQA is smaller than the scope of VQAv2. However, this is a deliberate choice: Evaluating with a large scope of mixed questions makes it difficult to find the exact strengths of models quantitatively. Our benchmark aims to provide a clear and constrained environment to evaluate specific capabilities of VQA systems: object, activity, and attribute recognition.
>
> Figure 1 (a) shows our decision to crop the relevant object in ImageNet. We ask “What’s this?”, and the model responds with “A tree with no leaves” which is a reasonably good answer given the image-question pair. Even though the desired label is “porcupine”, the model gets distracted by other objects in the image. In a standard classification setup this is less of a problem, since the label space reduces the ambiguity of the answer. To stay with the example of Figure 1 (a), tree is not a category of ImageNet; therefore, a classification or retrieval model cannot answer with a tree class. On the contrary, in our open-ended setup, the model must answer the question in a free manner without knowing the label space, increasing the ambiguity when the image contains more than a single object. By focusing on a cropped object of interest, we effectively reduce the ambiguity of the answer, allowing for a more precise assessment of the model’s ability to recognize and classify the specific object. To allow comparison with other works that test on ImageNet, we also provide results on ImageNet with full images in Supp. Tab. 4 and 5.
>
> In general, our scope of VQA includes asking about objects, actions and attributes. We acknowledge that there are many other facets of VQA systems that merit exploration, such as model performance on natural VQA datasets like VQAv2 and VizWiz. The scope of our benchmark could certainly be increased in future work e.g. by asking about object counts, spatial relations, compositional reasoning (e.g. having multiple object-attribute pairs in the same image and asking about their relations), testing generalization to rare scenes, or other scenarios. We believe our benchmark design and analysis of metrics will pave the way to test Text-VLMs on such abilities.
>
>
> **Answer to question 2**
>
> When developing ActivityNet-oVQA, we decided using only a single image, since the purpose of the whole benchmark is to evaluate models which answer questions on images. While extending this to VideoQA models is certainly a possible future direction, in this work our focus remains on images. Consistent with standard practice in the literature, we chose the middle frame of each segment as our input. Supp. Fig. 3 shows 4 example scenes which can all be answered given the middle frame shown in the figure.
>
> To determine the effectiveness of selecting the middle frame among all others, we conduct an ablation study, testing all models over nine frame selections by dividing the segment into nine parts and selecting the closest frame. The results are shown in Supp. Fig. 10. We measure accuracy as the average response to the same three initial questions across all models. The results suggest that frame selection does not have a significant impact on performance, especially since frames are selected from short segments in which the relevant action occurs.
> Additionally, our choice of using ActivityNet as a VQA dataset is motivated by its well-defined set of activities within a semantic hierarchy which allow us to ask follow-up questions about fine-grained activities.

---

### Author Response · Authors · 2023-11-18

Dear reviewers,

We thank you for taking the time to review our paper. Your insightful comments help to improve our work.

We have revised our submission, marked all content changes in blue, and uploaded a new version of the main document and supplementary. We made minor adjustments to the model output postprocessing as described at the end of Supp. Section A1. Consequently, we recomputed model performances, updated our tables, and highlighted the changes. These performance changes are minor and do not affect the ranking between models. Also, we added the model output length to Supp. Table 6 to help show the effect of hyperparameter choices for text generation. Finally, we made quality improvements like fixing typos and improving the typesetting.

For all other updates kindly refer to the answers we post to individual reviewers.

---

### Meta-Review · Area_Chair_jbej · 2023-12-14

**Metareview:**

This paper proposes a novel benchmark for evaluating Visual Question Answering (VQA) models, focusing on their ability to recognize fine-grained objects, activities, and attributes. The reviewers largely agree that the proposed benchmark addresses important limitations of existing evaluations and offers a valuable tool for assessing VQA model capabilities.

Some reviewers raise concerns about the benchmark's focus on cropped images and Imagenet data, potentially limiting its applicability to natural use cases with full, uncropped images. Please supplement information regarding the oVQA dataset, such as question/answer lengths, label distribution, and question formats, is missing, raising potential concerns about dataset balance and evaluation quality.

**Justification For Why Not Higher Score:**

Good and solid effort and valuable contribution to VQA evaluation. But, it is not ground-breaking level.

**Justification For Why Not Lower Score:**

While some concerns regarding data details and image space limitations require further attention, the paper's strengths outweigh its weaknesses. The authors can further strengthen their paper and its potential impact on the VQA research community by addressing points above, mainly clarification and better presentation.

---

### Decision · Program_Chairs · 2024-01-16

Accept (spotlight)